# Agonist-induced dimer dissociation as a macromolecular step in G protein-coupled receptor signaling

Julian Petersen [1], Shane C. Wright[1], David Rodríguez[2], Pierre Matricon[3], Noa Lahav[4], Aviv Vromen[4], Assaf Friedler[4], Johan Strömqvist[5], Stefan Wennmalm[6], Jens Carlsson[3] & Gunnar Schulte[1,7]

G protein-coupled receptors (GPCRs) constitute the largest family of cell surface receptors. They can exist and act as dimers, but the requirement of dimers for agonist-induced signal initiation and structural dynamics remains largely unknown. Frizzled 6 ($FZD_6$) is a member of Class F GPCRs, which bind WNT proteins to initiate signaling. Here, we show that $FZD_6$ dimerizes and that the dimer interface of $FZD_6$ is formed by the transmembrane α-helices four and five. Most importantly, we present the agonist-induced dissociation/re-association of a GPCR dimer through the use of live cell imaging techniques. Further analysis of a dimerization-impaired $FZD_6$ mutant indicates that dimer dissociation is an integral part of $FZD_6$ signaling to extracellular signal-regulated kinases1/2. The discovery of agonist-dependent dynamics of dimers as an intrinsic process of receptor activation extends our understanding of Class F and other dimerizing GPCRs, offering novel targets for dimer-interfering small molecules.

[1] Department of Physiology & Pharmacology, Section for Receptor Biology & Signaling, Karolinska Institutet, SE-171 77 Stockholm, Sweden. [2] Science for Life Laboratory, Department of Biochemistry and Biophysics, Stockholm University, SE-106 91 Stockholm, Sweden. [3] Science for Life Laboratory, Department of Cell and Molecular Biology, Uppsala University, BMC, Box 596, SE-751 24 Uppsala, Sweden. [4] Institute of Chemistry, The Hebrew University of Jerusalem, Givat Ram, Jerusalem 91904, Israel. [5] Single Technologies AB, SE-114 28 Stockholm, Sweden. [6] Department of Applied Physics, Experimental Biomolecular Physics group, Science for Life Laboratory, Royal Institute of Technology, Solna SE-171 65, Sweden. [7] Department of Experimental Biology, Faculty of Science, Masaryk University, Kotlarska 2, 61137 Brno, Czech Republic. Julian Petersen and Shane C. Wright contributed equally to this work. Correspondence and requests for materials should be addressed to G.S. (email: gunnar.schulte@ki.se)

The superfamily of G protein-coupled receptors (GPCRs; a list of abbreviations used in this work is presented in Supplementary Note 1) mediates the effects of a plethora of endogenous and exogenous substances such as small molecules, peptides, proteins, lipids, ions, and odorants, and offers efficient targets for drug treatment[1, 2]. According to sequence homology, GPCRs were grouped into Classes A, B, C, F (Frizzleds), adhesion receptors, and other 7 transmembrane (TM) spanning receptors[3]. In addition to the well-understood signaling unit of a monomeric GPCR[4], it has been demonstrated that GPCRs can exist as homomeric and heteromeric dimers across the different classes of the GPCR superfamily[5, 6]. Even though the existence of GPCR homomeric and heterodimers is generally accepted, our understanding of their role in receptor function and signal initiation is very limited[6, 7].

Class F receptors are classified as members of the GPCR superfamily based on structural and functional resemblance to class A, B, and C family receptors[8]. Frizzleds (FZDs) regulate a number of processes during embryonic development, stem cell regulation, and adult tissue homeostasis. Deregulation or misexpression of FZDs leads to pathogenesis, including, but not limited to, cancer and neurologic disorders; thus, making them attractive drug targets[8, 9]. In mammals, there are 10 Frizzleds ($FZD_{1-10}$), which are activated by the WNT family of lipoglyco-proteins through interaction with the N-terminal cysteine-rich domain (CRD) of FZD[10, 11]. Little is known about receptor complex constitution with regard to the stoichiometry of WNT to FZD and FZD to intracellular mediators, such as the phospho-protein Disheveled (DVL) and heterotrimeric G proteins[12, 13]. Nevertheless, it has been shown that $FZD_{1,2,3,4}$ dimerize and that dimerization has implications for signaling[14, 15].

Based on live cell imaging experiments, we provide evidence that $FZD_6$ dimerizes and that the dimer undergoes WNT-5A-induced dissociation and re-association. Employing mutational analysis and biochemical approaches, we map the $FZD_6$–$FZD_6$ dimer interface to TM4 and TM5—findings that are supported by atomic resolution receptor models. In addition, expression of peptides interfering with $FZD_6$ dimerization in mouse lung epithelial (MLE-12) cells endogenously expressing $FZD_6$ reduces basal extracellular signal-regulated kinase 1/2 (ERK1/2) phosphorylation in like manner to cells lacking a functional copy of $FZD_6$. This, in combination with data pointing to the enhanced signaling capacity of the $FZD_6$ dimer mutant in the absence of ligand, argues that $FZD_6$ dimers are crucial in the establishment of functional inactive-state complexes and that monomeric $FZD_6$ promotes signaling to ERK1/2. Finally, our results suggest that the $FZD_6$ monomer, and not the dimer, is the minimal signaling unit supporting the idea that receptor dissociation precedes signal initiation.

## Results

**$FZD_6$ forms homodimers.** Previously, we reported that $FZD_6$ acts in complex with the phosphoprotein DVL and heterotrimeric $G\alpha_{i1}$ or $G\alpha_q$ proteins[16]. Interestingly, interaction of the heterotrimeric G proteins $G\alpha_{i1}$ and $G\alpha_q$ with $FZD_6$ relied on balanced levels of DVL, which interacts with the conserved C terminal KTxxxW sequence and the third intracellular loop of FZDs[8, 17, 18]. The G protein-receptor interface of many GPCRs largely coincides with the interface that mediates FZD–DVL interaction. Since simultaneous interaction of DVL and G proteins with $FZD_6$ appears competitive, we hypothesize that $FZD_6$ forms a dimeric quaternary structure, in which one protomer interacts with DVL and the other one with the heterotrimeric G protein.

To assess receptor–receptor interaction, we used dual color fluorescence recovery after photobleaching (dcFRAP)[16, 19, 20].

HEK293 cells were transiently transfected with V5-$FZD_6$-mCherry and $FZD_6$-GFP (Fig. 1a). Surface immobilization of the V5-$FZD_6$-mCherry was achieved with a biotinylated anti-V5 antibody and avidin[19] and routinely resulted in a dramatic reduction in the recoverable fluorescence of V5-$FZD_6$-mCherry defining the mobile fraction of the protein of interest. As predicted, the physical interaction between the V5-$FZD_6$-mCherry and $FZD_6$-GFP resulted in a markedly reduced mobile fraction of $FZD_6$-GFP upon biotinylated anti-V5 antibody/avidin crosslinking. Crosslinking reduced the basal mobile fraction ($80.8 \pm 2.0\%$; mean $\pm$ standard error of the mean, s.e.m.) of $FZD_6$-GFP to $70.7 \pm 2.4\%$ arguing for the presence of a higher order complex such as a receptor dimer ($P = 0.0045$; $t = 2.975$; df $= 50$; two-tailed $t$-test) (Fig. 1b–d). Importantly, antibody-treated cells co-expressing V5-$FZD_6$-mCherry and cannabinoid $CB_1$ receptor-GFP ($CB_1$-GFP), an unrelated Class A GPCR, did not exhibit a reduction in the mobile fraction of $CB_1$-GFP upon crosslinking, underlining the specificity of the $FZD_6$–$FZD_6$ interaction and of this technique (Fig. 1e–g). Direct statistical comparison (one-way ANOVA) of the CL-induced differences in the mobile fraction of the respective GFP constructs in the conditions V5-$FZD_6$-mCherry/$FZD_6$-GFP and V5-$FZD_6$-mCherry/$CB_1$-GFP indicate that the observed interactions are specific (Supplementary Fig. 1a). In order to detect protein–protein interactions by dcFRAP, the relative expression levels of the crosslinked receptor population (mCherry) must be in excess relative to the mobile receptor population (GFP)[21]. As a means of estimating stoichiometry, we designed a TM protein carrying an extracellular mCherry and an intracellular GFP (mCherry–TM–GFP), which we imaged with identical acquisition settings to our dcFRAP set-up over a range of fluorescence intensity levels. Values were fit to a linear regression providing a fluorescence intensity ratio of GFP to mCherry of $1.6 \pm 0.1$ ($R^2 = 0.8739$; $N = 54$) (Supplementary Fig. 1b). Throughout the manuscript, only values with an intensity(GFP)/intensity (mCherry) $<1.6$ for a fixed laser ratio 488/543 were included in the analysis of dcFRAP experiments.

We also performed ratiometric Förster resonance energy transfer (FRET) measurements using the same C-terminally tagged $FZD_6$ constructs employed in dcFRAP experiments; however, we were not able to detect energy transfer between the fluorophores possibly due to suboptimal fluorophore orientation at the 210 aa long C-terminal tail.

Next, we set out to corroborate the $FZD_6$–$FZD_6$ interaction by employing fluorescence cross-correlation spectroscopy (FCCS), which allowed for analysis of low receptor expression levels (from 10–1000 receptors per $\mu m^2$ in each examined HEK293 cell). Due to its single-molecule sensitivity, FCCS can be performed using low levels of protein[22] and without the need for crosslinking. FCCS measurements were performed on HEK293 cells co-expressing $FZD_6$-GFP and V5-$FZD_6$-mCherry, and a prominent cross-correlation amplitude was observed (Fig. 1h–j). Analysis of crosstalk-corrected FCCS data obtained in $FZD_6$-GFP/$FZD_6$-mCherry-transfected cells is in agreement with a model assuming that all $FZD_6$ exists as dimers (RMS = 0.12) (either as $FZD_6$-GFP/$FZD_6$-GFP, $FZD_6$-GFP/V5-$FZD_6$-mCherry or V5-$FZD_6$-mCherry/V5-$FZD_6$-mCherry dimers). The crosstalk correction in the model assumed that $CB_1$-mCherry was monomeric (Fig. 1j).

**TM4 and TM5 form the interface for $FZD_6$ dimerization.** To further support our findings of a potential $FZD_6$ dimer, we generated a $FZD_6$ homology model using the crystal structure of Smoothened (SMO), which crystallizes as a parallel homodimer and has ~30% sequence identity to $FZD_6$ in the TM region[23]

(Supplementary Fig. 2). Accordingly, the interface of the $FZD_6$ dimer was predicted to involve the fourth and fifth TM helices (denoted TM4 and TM5)[23]. Access to this recently determined structure allowed us to generate atomic resolution models of the $FZD_6$ homodimer, suggesting extensive contacts between the protomers and an interface with a buried surface area (BSA) of ~1150 Å². The principal contributing force to the interaction is predicted to originate from van der Waals contacts between the extracellular halves of TM4 and TM5 (Fig. 2a–d; Supplementary Fig. 3). To further support the likelihood of the predicted dimer interface, molecular dynamics (MD) simulations of the dimer model in a hydrated lipid bilayer were performed (Supplementary

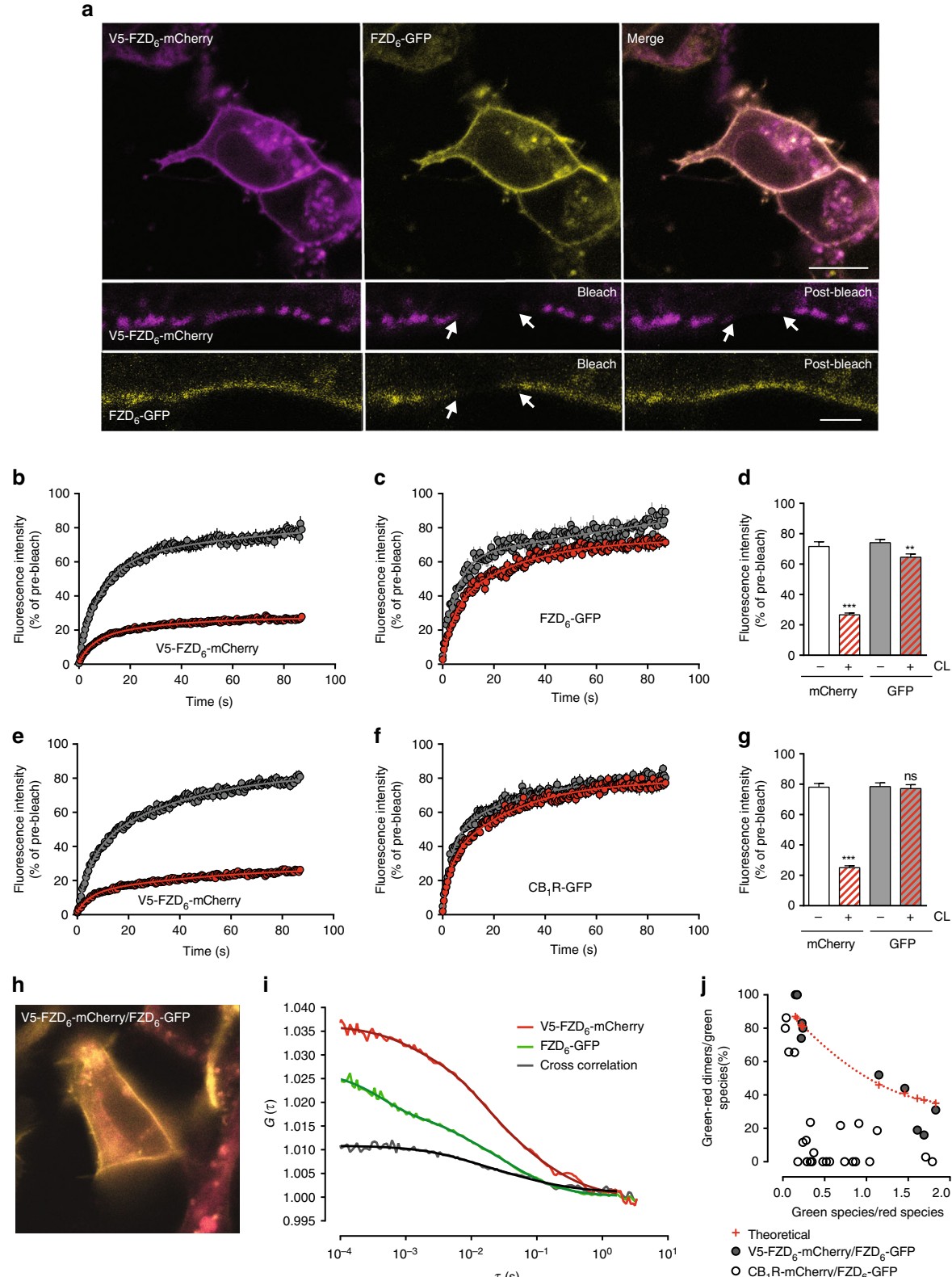

Fig. 2c–d). Three independent calculations were carried out for the system, totaling 300 ns of unrestrained simulation time. After an initial equilibration phase, which involved only minor rearrangements in the dimer interface, the system remained stable for the remainder of the simulation (Supplementary Fig. 2d).

In order to determine whether this interface is required for the dimerization of $FZD_6$ in living cells, we systematically introduced a number of mutations into the extracellular parts of TM4 and TM5 (Supplementary Fig. 4). $Y369^{5.45f}$ was predicted to be in the core of the dimer interface and part of a cluster of hydrophobic residues (e.g. $F337^{4.55f}$, $V340^{4.58f}$, and $M341^{4.59f}$). The ionizable residues $D365^{5.41f}$ and $R368^{5.44f}$ were typically involved in interactions with $N346^{4.64f}$ and $A344^{4.62f}$ on the other protomer. Mutation of $D365^{5.41f}$, $R368^{5.44f}$, and $Y369^{5.45f}$ to alanine was predicted to disrupt favorable interactions in this region, thereby impairing $FZD_6$ homodimerization. As surmised, dcFRAP experiments with the D365A, R368A, Y369A triple mutant (hereafter called the $FZD_6$ dimer mutant) of $FZD_6$-GFP in combination with V5-$FZD_6$-mCherry did not result in a decrease of the mobile fraction of $FZD_6$-GFP upon V5 antibody/avidin-mediated crosslinking, supporting the importance of these residues for dimer formation ($P = 0.1438$; $t = 1.482$; $df = 57$; two-tailed $t$-test) (Figs. 2e and f). The absence of a CL-induced reduction in the mobile phase of the GFP-tagged dimer mutant $FZD_6$ also supports the existence of TM4/5 interface dimers rather than higher oligomeric structures, which would in part depend on other receptor–receptor interfaces. Interestingly, exchanging these residues for chemically conserved amino acids (D365E, R368K, Y369F—hereafter called the $FZD_6$ dimer control mutant) maintained the intrinsic ability of $FZD_6$ to form dimers ($P \leq 0.0001$; $t = 4.839$; $df = 153$; two-tailed $t$-test) (Figs. 2g and h). Further mutational analysis, including the Y369A mutation on TM5 and additional residues located on TM4 (F337, M341, M345, N346), is shown in Supplementary Fig. 4. The $FZD_6$ F337A mutant exhibited a reduced ability to dimerize, further supporting the involvement of TM4 in the dimer interface.

Based on the hypothesis that a larger proportion of the $FZD_6$ wt population exists as dimers compared to the receptor population of $FZD_6$ dimer mutant, we measured the diffusion of the two receptor variants in living HEK293 cells using fluorescence correlation spectroscopy (FCS) measurements. Employing GFP-tagged $FZD_6$ and the $FZD_6$-GFP dimer mutant, it was possible to determine the transit time ($\tau_D$) of the molecules passing through the detector volume as a factor of molecular size and speed (Fig. 2i). The substantially lower values of $\tau_D$ in the case of the dimer mutant ($30.1 \pm 2.0$ ms) compared to wt

($52.4 \pm 4.1$ ms) provides strong evidence for higher order complexes in the case of wt $FZD_6$ ($P < 0.0001$; $t = 4.552$; $df = 29$; two-tailed $t$-test). In order to reduce potential variability in diffusion speed of the receptor species caused by binding of endogenously expressed WNTs, we pretreated the cells with 5 µM C59, a porcupine inhibitor, which blocks secretion of WNT proteins[24].

Furthermore, we employed protein biochemical analysis of synthetic TM4 and TM5 peptides to validate the involvement of TM helix 4 and 5 of $FZD_6$ in receptor dimerization. Analytical size-exclusion chromatography (SEC) experiments using synthetic TM4 and TM5 peptides in $n$-Dodecyl β-D-maltoside (DDM) showed that the complex of TM4 and TM5 eluted at a greater volume than TM4 and TM5 alone indicative of binding between the two peptides as well as the adoption of a more compact structure than either of the peptides alone. On the other hand, the combination of TM4 and TM5 (D365A, R368A, Y369A) mutant peptide (Fig. 3b) did not result in a shift to a higher elution volume suggestive of a weaker complex. Thus, we conclude that the physical interaction observed in TM4/5 peptides was lost or weakened in the D365A-, R368A-, Y369A-mutated peptide of TM5 (Fig. 3a and b) highlighting the intrinsic importance of $D365^{5.41f}$, $R368^{5.44f}$, $Y369^{5.45f}$ for $FZD_6$ dimer formation. To confirm the secondary structure of the synthetic peptides in solution, circular dichroism (CD) experiments employing purified TM4 and TM5 peptides in detergent solution (DDM) revealed that the TM peptides became structured and α-helical in a lipophilic environment (Supplementary Fig. 5a–d). In addition, isothermal titration calorimetry (ITC) showed that purified TM4 and TM5 peptides interact with micromolar affinity (Fig. 3c), whereas TM4 does not interact with the mutant TM5 (Fig. 3d). Control experiments were conducted by titrating 820 µM TM5 or TM5 mut into triple distilled water (TDW) with 0.5% DDM, and TDW, 0.5% DDM into 165 µM TM4 (Supplementary Fig. 5e–g). These experiments confirmed that the heat was produced from binding and not from the peptide dilution or conformational change. The reductionist approach of using purified TM peptides in detergent solution validates the role and contribution of TM4/5 for the receptor interface and underlines the importance of the TM4/5 interaction involving $D365^{5.41f}$, $R368^{5.44f}$, and $Y369^{5.45f}$.

**Role of intracellular components for $FZD_6$ dimerization.** GPCR dimer formation is seen as a receptor-intrinsic process and the requirement for intracellular receptor-binding proteins that

---

**Fig. 1** $FZD_6$ forms dimers. **a** HEK293 cells expressing V5-$FZD_6$-mCherry and $FZD_6$-GFP were used for dcFRAP. Arrows depict a ROI of 1.80 × 1.80 µm that was photobleached. Size bar, 10 µm. Lower panels provide fluorescence prior (*left*), directly after (bleach), and 100 s after (post-bleach) photobleaching. Size bar–2 µm. **b**, **c** Curves provide fluorescence intensity data from dcFRAP in the absence (*gray*) and presence (*red*) of biotinylated anti-V5/avidin crosslinking (CL) for V5-$FZD_6$-mCherry and $FZD_6$-GFP. The CL-induced decrease in the mobile fraction of $FZD_6$-GFP indicates receptor–receptor interaction. **d** Bar graph shows fluorescence intensity averages of the mobile fraction across different conditions within the time frame of 85–101 s (including 15 s pre-bleach measurements). *White bars*—V5-$FZD_6$-mCherry; *gray bars*—$FZD_6$-GFP; hatched—CL. $N = 21$ ROIs before CL; $N = 31$ after CL from three independent experiments. $P = 0.0045$; $t = 2.975$; $df = 50$. **e**–**g** Corresponding negative control dcFRAP experiments for $FZD_6$ interaction employing V5-$FZD_6$-mCherry and $CB_1$-GFP. $P = 0.7030$; $t = 0.3831$; $df = 61$. $N = 29$ ROIs before CL; $N = 34$ ROIs after CL from three independent experiments. Error bars provide standard error of the mean (s.e.m.). **$P < 0.01$; *ns* non significant (two-tailed $t$-test). For a statistical comparison of the $FZD_6$/$FZD_6$ and $FZD_6$/$CB_1$ data sets presented in **d** and **g**, see Supplementary Fig. 1a. Only ROIs wherein the crosslinked receptor population was in excess to the uncrosslinked were included in the data analysis (see Supplementary Fig. 1b). **h** HEK293 cells expressing V5-$FZD_6$-mCherry/$FZD_6$-GFP used for FCCS. **i** FCCS amplitudes from a representative experiment are shown. Autocorrelations in *green*—$FZD_6$-GFP and *red*—V5-$FZD_6$-mCherry; cross-correlation amplitude in black. $G(\tau)$ provides the fluorescence correlation amplitude. $\tau/s$ gives time shift per s. **j** The scatter plot summarizes FCCS values obtained in cells co-expressing $FZD_6$-GFP/V5-$FZD_6$-mCherry (*filled circles*) or cannabinoid $CB_1$ receptor ($CB_1$-GFP)/V5-$FZD_6$-mCherry (*open circles*). Data presentation shows the percentage of dimerization over varying numbers of *green* over *red* particles. The crosses/dotted line represent a theoretical model assuming 100% $FZD_6$ dimerization. The crosstalk correction in the model assumed that $CB_1$-mCherry was monomeric. $N = 10$ cells expressing $FZD_6$-GFP/V5-$FZD_6$-mCherry; $N = 25$ cells expressing $CB_1$-GFP/V5-$FZD_6$-mCherry from 6 independent experiments. Data are based on two to eight 10 s acquisition periods per cell (see also Supplementary Fig. 1b)

facilitate dimerization has not been addressed. $FZD_6$ interacts with heterotrimeric G proteins and the phosphoprotein DVL, where DVL regulates $FZD_6$ interaction with $G\alpha_{i1}/G\alpha_q$[16]. In order to elucidate the involvement of DVL and heterotrimeric G proteins in $FZD_6$–$FZD_6$ interaction, we performed dcFRAP experiments (V5-$FZD_6$-mCherry/$FZD_6$-GFP) in cells with either decreased or increased expression levels of DVL. In contrast to previous findings where varying levels of DVL regulated $FZD_6$-G

protein assembly, neither depletion ($P \leq 0.0001$; $t = 6.854$; df = 53; two-tailed $t$-test) nor excess of DVL ($P \leq 0.0001$; $t = 6.243$; df = 77; two-tailed $t$-test) abrogated $FZD_6$ dimer formation (Fig. 4a–d)[16]. Interestingly, the $FZD_6$ dimer mutant retained the ability to interact with the scaffold protein DVL, but not with $G\alpha_{i1}$ as assessed by dcFRAP between the $FZD_6$ dimer mutant and DVL2-GFP ($P < 0.0001$; $t = 5.173$; df = 74; two-tailed $t$-test) (Fig. 4e) or $G\alpha_{i1}$-GFP ($P = 0.5646$; $t = 0.5801$; df = 48;

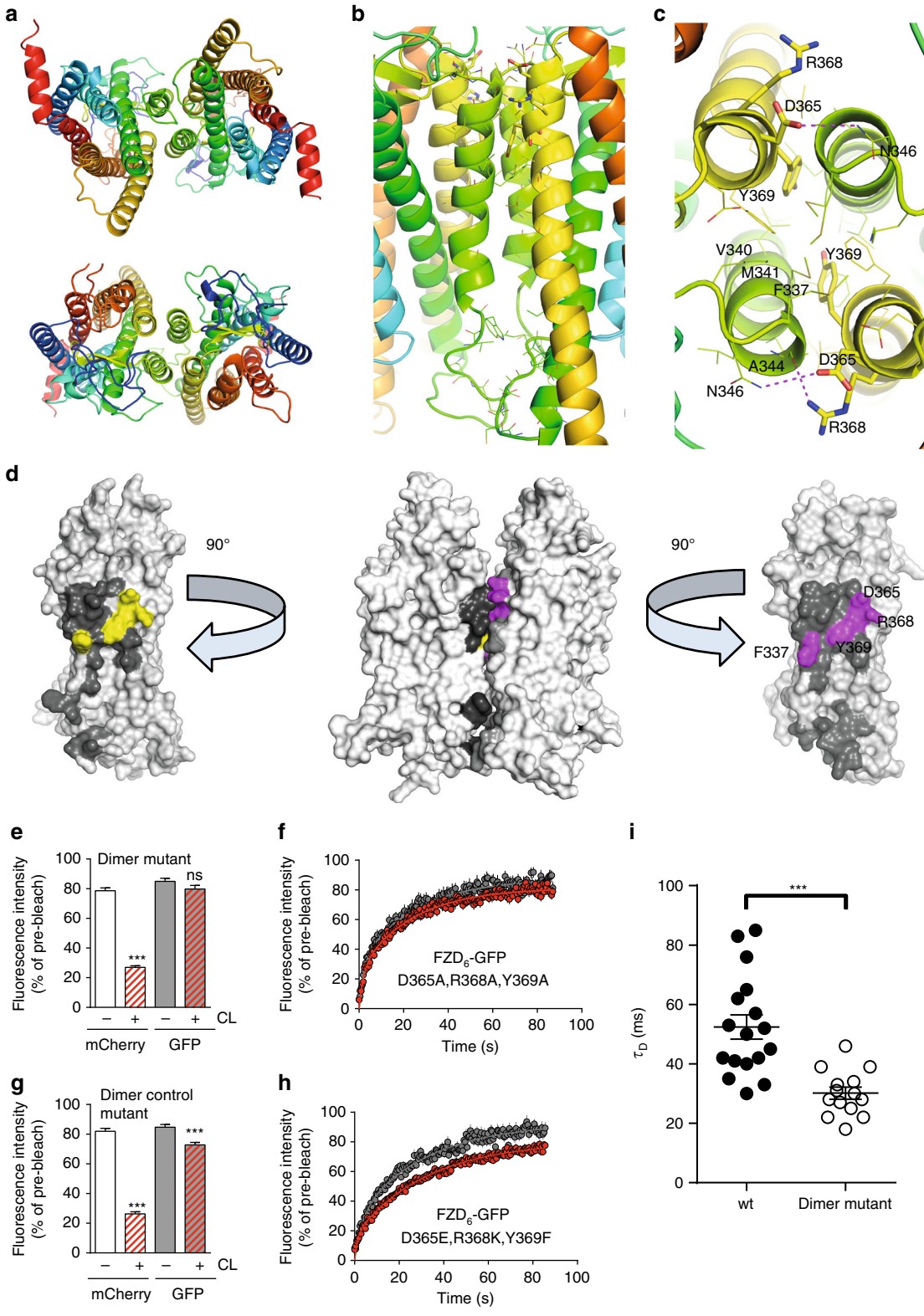

two-tailed $t$-test) (Fig. 4f). To further elucidate the mechanism of dimerization, we took advantage of the differential cellular localization of both wt $FZD_6$ and the DVL-biased $FZD_6$-R511C nail dysplasia mutant that does not assemble with $G\alpha_{i1}/G\alpha_q$ proteins[16, 25] (Fig. 4g). While co-expression of $FZD_6$-mCherry and $FZD_6$-GFP resulted in distinct cell surface expression of both receptor populations, co-expression of $FZD_6$-R511C-GFP together with $FZD_6$-mCherry resulted in predominant intracellular localization of both constructs. $FZD_6$-mCherry was trapped intracellularly due to the strong mislocalization phenotype of the mutant[25] and dimerization between wt and $FZD_6$ R511C. On the other hand, the $FZD_6$-GFP dimer mutant maintained cell surface localization despite co-expression with $FZD_6$-R511C-mCherry. Since neither $FZD_6$ R511C[16] (able to dimerize) nor the dimer mutant of $FZD_6$ (not able to dimerize) were able to assemble with heterotrimeric G proteins, we conclude that neither DVL nor G protein binding are required for $FZD_6$ dimer formation (Fig. 4g).

**Dynamic $FZD_6$ dimerization.** Currently, little is known about FZD complex composition, dynamics, stoichiometry, and the contribution of DVL and heterotrimeric G proteins. It was previously demonstrated that WNT-5A, an agonist for FZDs, induced the dissociation of heterotrimeric G proteins $G\alpha_{i1}$ and $G\alpha_q$ from the $FZD_6$ inactive-state complex[16]. Based on these findings, the question arose if not only the receptor-G protein, but also the receptor–receptor interaction could be ligand sensitive. To this end, dcFRAP assays in cells expressing V5-$FZD_6$-mCherry and $FZD_6$-GFP were performed with or without WNT-5A stimulation (300 ng ml$^{-1}$) for 5, 10, 15, and 20 min. While interaction of V5-$FZD_6$-mCherry and $FZD_6$-GFP was measured at baseline (mobile fraction: $64.6 \pm 2.1\%$), the addition of WNT-5A led to a transient loss of CL-induced decrease in the $FZD_6$-GFP mobile fraction at the first measured time point ($71.8 \pm 3.2\%$ at 5 min). Subsequently, the mobile fraction of the receptor returned to values resembling unstimulated ($61.0 \pm 2.2\%$ at 20 min) ($P = 0.0219$; F (5, 109) = 2.757; one-way ANOVA) (Fig. 5a, b). Since these experiments, which point to a decrease of higher order complexes in the total receptor population, were performed with a relatively high degree of receptor overexpression and receptor crosslinking, we aimed to create conditions with low expression levels of $FZD_6$ with unimpeded mobility. FCCS experiments on HEK293 cells expressing V5-$FZD_6$-mCherry and $FZD_6$-GFP supported the dynamics and kinetics of the agonist-induced dissociation/re-association of the $FZD_6$ dimer ($P = 0.0042$; F (7, 39) = 3.628; one-way ANOVA) (Fig. 5c–e). Cross-correlation of mCherry/GFP fluorescence indicated a stable baseline of *red/green* dimers, after which a transient decrease in $FZD_6$–$FZD_6$ interaction could be detected upon stimulation with WNT-5A (300 ng ml$^{-1}$) (Fig. 5c–e). Similar to the kinetics of the dcFRAP analysis,

*red/green* correlation (Ngr/Nr values in Fig. 5c; $G_{cc}$ values in Fig. 5d) returned to values resembling re-association of the $FZD_6$ promoters in the range of minutes. In order to rule out receptor internalization as a confounding factor for the dissociation observed in the FCCS experiments, we quantified the number of *red* (Nr) and *green* (Ng) fluorescent particles over the course of dynamic FCCS measurements (Fig. 5c). The transient increase in particle number and the return to values resembling prestimulation levels (values are not corrected for bleaching) support the idea of agonist-induced complex dynamics instead of agonist-induced receptor trafficking. To corroborate this finding, we assessed the degree of $FZD_6$ internalization after exposure to WNT-5A by confocal microscopy (Fig. 5f). This was achieved by pulsing HEK293 cells expressing SNAP-$FZD_6$ with a cell-impermeable SNAP substrate. Within the time frame of the FCCS experiments, minimal internalization of SNAP-$FZD_6$ was observed and this was not enhanced by the addition of recombinant WNT-5A (similar to what had previously been reported using the $FZD_6$-mCherry construct[26]). For the purpose of comparison, 10 μM isoproterenol stimulation of SNAP-tagged $\beta_2$-adrenergic receptors, a Class A GPCR known to undergo rapid receptor-mediated endocytosis, resulted in the appearance of numerous endosome-like structures after 15 min, similar to what had been described by others[27]. Based on dcFRAP and FCCS experiments, we propose the model shown in Fig. 5g.

**Modulation of $FZD_6$ dimers and downstream signaling.** In order to translate our molecular and mechanistic findings for the importance of $FZD_6$ dimers at the cellular level, we turned to MLE-12 cells. $FZD_6$ is highly expressed in adult and fetal human lung tissue, but its function in the lung is mostly unknown[28, 29]. Immunohistochemistry data from the Human Protein Atlas indicate strong membranous $FZD_6$ staining in human bronchial epithelial cells[30] (http://www.proteinatlas.org/ENSG00000164930-FZD6/tissue/bronchus). Cells derived from the distal bronchiolar and alveolar epithelium of SV40 transgenic mice, otherwise known as murine lung epithelial (MLE-12) cells[31], express $FZD_6$ and to a lesser extent $FZD_2$ at the mRNA level as shown by quantitative PCR (Supplementary Fig. 6). On the protein level, $FZD_6$ is readily detectable in cell lysates indicating that the receptor is expressed in sufficient amounts to be of functional relevance in these cells while providing a suitable model to support the role of $FZD_6$ dimerization at endogenous receptor expression levels. Molecular validation of the cell system indicated that the addition of recombinant and purified WNT-5A (300 ng ml$^{-1}$) led to a time-dependent, partially pertussis toxin-sensitive phosphorylation of ERK1/2, with maximal increase in P-ERK1/2 at 5 min (Fig. 6a, b; Supporting Fig. 6). Interestingly, the kinetics of WNT-5A-induced ERK1/2 phosphorylation matched the time course of $FZD_6$ de- and

**Fig. 2** $FZD_6$ dimer interface is formed by TM4 and TM5. **a–c** Presents the homology model of the $FZD_6$ homodimer interactions involving residues F337$^{4.55f}$, A344$^{4.62f}$, N346$^{4.64f}$, D365$^{5.41f}$, R368$^{5.44f}$ and Y369$^{5.45f}$. Rainbow coloring, N-terminus—*blue*; C-terminus—*red*. Main interactions involve TM4 (*green*) and TM5 (*yellow*). **a** Upper panel: from intracellular side; *lower panel*: from extracellular side. **b** side view and **c** extracellular view with zoom on dimer interface. Residues are shown as sticks in **b** and **c**. **d** Space filling model of the $FZD_6$ dimer and each monomer (side view). Dimer interface area (ca 1,150 Å$^2$) is colored in *dark gray*. *Yellow/magenta* identify residues important for dimer interaction as demonstrated by site-directed mutagenesis. For more details, see Supplementary Figs. 2 and 3. **e, f** dcFRAP analysis of V5-$FZD_6$-mCherry and the $FZD_6$-GFP dimer mutant showed that the mobile fraction of the dimer mutant is not affected by biotin-anti-V5/avidin surface crosslinking indicating a reduced ability to form dimers. Only data for $FZD_6$-GFP in the absence (*gray*) and presence (*red*) of CL are shown. $P = 0.1438$; $t = 1.482$; df = 57. N = 23 ROIs before CL; N = 36 after CL from three independent experiments. For analysis of additional mutations see Supplementary Fig. 4. **g, h** A corresponding $FZD_6$-GFP dimer control mutant, where mutations preserving side chain chemical properties were introduced, still interacts with the wt V5-$FZD_6$-mCherry as analyzed by dcFRAP. $P < 0.0001$; $t = 4.839$; df = 153. N = 75 ROIs before CL; N = 80 ROIs after CL from three independent experiments. **i** FCS experiments measuring the translational diffusion time ($\tau_D$) of $FZD_6$-GFP and the dimer mutant $FZD_6$-GFP in the presence of the porcupine inhibitor C59 (5 μM, overnight). Data are presented as a scatter dot plot. $P < 0.0001$; $t = 4.552$; df = 29. N = 17 cells expressing $FZD_6$-GFP; N = 14 cells expressing $FZD_6$-GFP dimer mutant from three independent experiments. Error bar gives s.e.m. ***$P < 0.001$ (two-tailed $t$-test)

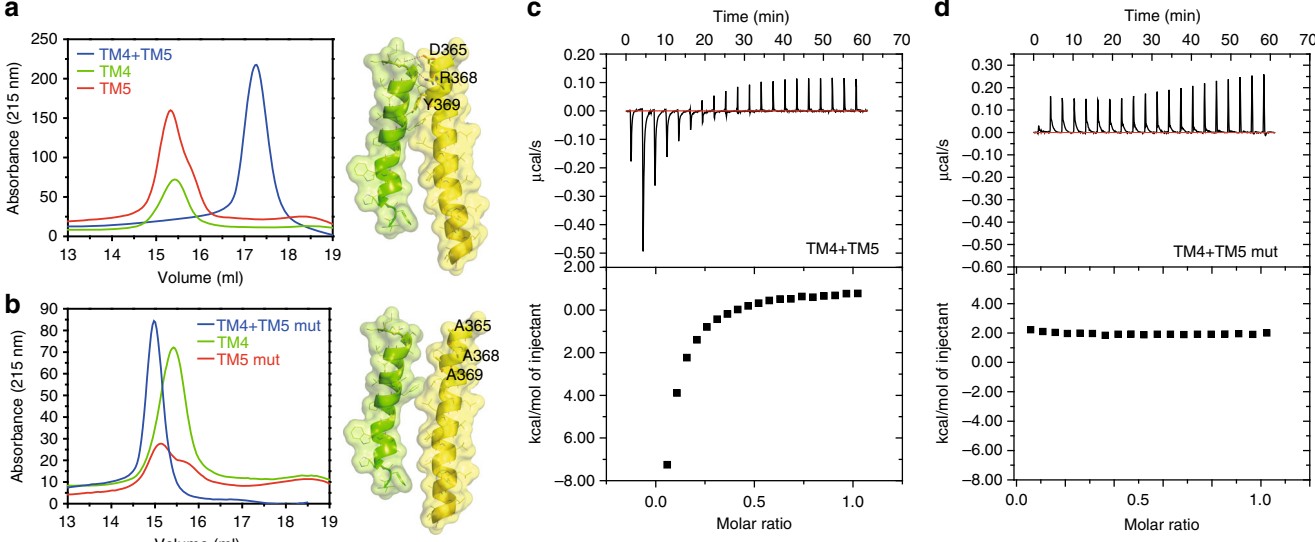

**Fig. 3** Mutation of TM5 decreases affinity between the TM4/5 peptides in DDM solution. **a, b** Size-exclusion chromatography experiments were performed in triple distilled water (TDW) and 0.01 % DDM. Final concentration of 500 µM from each peptide in TDW + 0.01 % DDM was injected into the size-exclusion column and their absorbance was detected as explained in the methods section. **a** Elution profiles of synthetic peptides TM4 and TM5 alone and in combination. **b** Elution profiles of TM4 and mutated TM5 (TM5 mut) peptides are shown. Helical peptide models (TM4—*yellow*; TM5—*green*) visualize the expected van der Waals interaction between TM4/5 and the lack thereof in the mutated peptide derived from the $FZD_6$ model presented in Fig. 2. **c** Isothermal titration calorimetry (ITC) experiments were performed in 0.5% DDM at 20 °C. ITC curves for the titration of 820 µM $FZD_6$-TM5 peptide into 165 µM $FZD_6$-TM4 are shown. **d** Titration of 820 µM $FZD_6$-TM5 mut peptide into 165 µM $FZD_6$-TM4 is shown. Fitting the binding curve to one set of sites resulted in an affinity (dissociation constant; $K_D$) in the micromolar range. The precise $K_D$ value could not be determined due to high background, which is likely to be caused by the dimerization of the $FZD_6$-derived peptides. Control experiments with TM4, TM5, and TM5 mut peptides alone are shown in Supplementary Fig. 5e–g

re-dimerization, suggesting a possible role for dimerization in signaling.

In order to functionally link ERK1/2 phosphorylation to $FZD_6$, we used CRISPR/Cas9-mediated gene editing to target $FZD_6$ in MLE-12 cells. Clonal analysis defined a heterozygous $FZD_6^{+/-}$ line, in which $FZD_6$ expression and cellular proliferation were visibly reduced (Fig. 6c; Supplementary Fig. 6). In addition, basal P-ERK1/2 levels were reduced in $FZD_6^{+/-}$ cells when compared to wt suggesting that downstream signaling to ERK1/2 depends on $FZD_6$ expression in MLE-12 cells (Fig. 6c). Along the same line, CRISPRa-induced enhancement of $FZD_6$ expression in MLE-12 cells resulted in increased C-RAF and ERK1/2 phosphorylation confirming that regulation of $FZD_6$ protein levels relates to the degree of ERK1/2 phosphorylation (Fig. 6d). Based on the $FZD_6$-mediated ERK1/2 phosphorylation, we were then able to investigate the role of $FZD_6$ dimers for signaling in MLE-12 cells using transient expression of the TM1/TM5 minigenes. For this purpose, we cloned hemagglutinin (HA)-tagged minigenes of TM1 (control) and TM5 (dimer-interfering) with an N-terminal membrane insertion sequence in order to express them as single membrane spanning peptides. Co-transfection of V5-$FZD_6$-mCherry and $FZD_6$-GFP with TM5 perturbed dimer formation, but not in the case of TM1, as assessed by dcFRAP ($P = 0.0472$; $t = 2.008$; df = 107; two-tailed $t$-test) (Supplementary Fig. 7a). As a means of controlling for membrane localization and proper orientation of the minigenes, we performed immunocytochemistry to detect the extracellular N-terminal HA-tag in fixed cells in the absence of detergent permeabilization (Supplementary Fig. 7b). Taking together the findings that the peptides are localized in the membrane, obtain the correct orientation and form helices in a lipophilic environment (Supplementary Fig. 5), supports our idea to use these minigenes as tools in cells to dissect the role of $FZD_6$ dimerization for ERK1/2 signaling in MLE-12 cells.

Overexpression of TM1, the minigene that does not affect $FZD_6$ dimerization in HEK293 cells, did not affect basal ERK1/2 levels in MLE-12 cells. On the other hand, when expressing the dimer-interfering TM5 minigene, basal P-ERK1/2 levels dropped to $56.7 \pm 11.4$ % ($P = 0.0091$; $t = 3.791$; df = 6; two-tailed $t$-test) (Supplementary Fig. 7c, d). We excluded off-target effects on other GPCRs by transiently transfecting either empty vector, TM1 or TM5 minigenes in HEK293 cells and stimulating with lysophosphatidic acid (10 µM; agonist at LPA receptors) and thrombin (10 U ml⁻¹; activating proteinase-activated receptors) as these are well-documented propagators of ERK1/2 signaling. We did not observe any difference in P-ERK1/2 induction in the presence of minigenes (Supplementary Fig. 7e). In short, we show that $FZD_6$ is required to maintain basal ERK1/2 signaling and that the dimer-interfering TM5 minigene impairs $FZD_6$ signaling. This raises the question as to how agonist-induced dimer dissociation and $FZD_6$ bound to TM5 peptide can have opposite signaling outcomes. To better understand the signaling capacity of wt and dimer mutant $FZD_6$, we overexpressed the two constructs in HEK293 cells and analyzed downstream ERK1/2 signaling. In serum-starved HEK293 cells, both the $FZD_6$ dimer mutant and $FZD_6$ wt increased basal P-ERK1/2 (Fig. 6e). Given the high degree of autocrine WNT signaling in various cell types, we used the porcupine inhibitor C59 (5 µM, overnight) to determine the effect of endogenous WNTs on receptor-mediated signaling. In other words, C59 could provide a tool for addressing the role of ligand-induced receptor dynamics for signal initiation. In the presence of C59, which abrogates endogenous WNT secretion[24], $FZD_6$ wt-induced levels of P-ERK1/2 were reduced to basal, whereas the $FZD_6$ dimer mutant maintained its ability to induce ERK1/2 phosphorylation (Fig. 6f, g). Furthermore, increasing expression of either $FZD_6$ wt or $FZD_6$ dimer mutant in C59-treated HEK293 cells strengthened the finding that the dimer mutant, but not the $FZD_6$ wt is capable of inducing P-ERK1/2 in

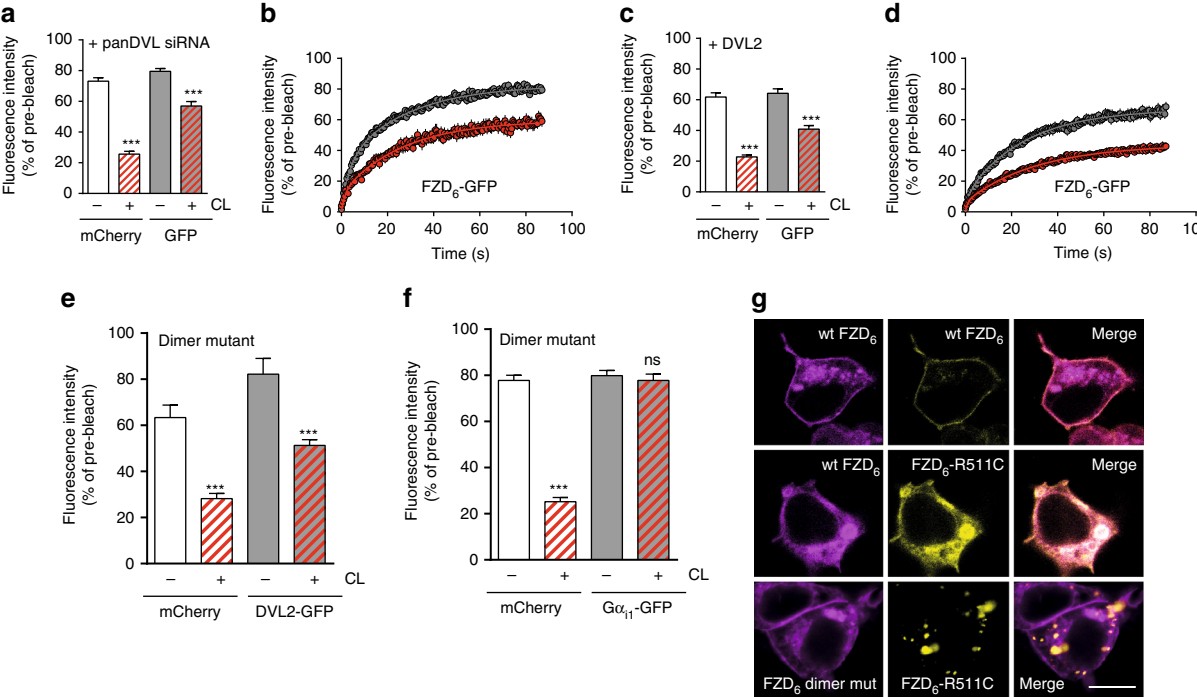

**Fig. 4** Absence of a role for DVL and G proteins in $FZD_6$ dimer formation. **a**–**d** Analogous to data shown in Fig. 1c, d, we investigated the effect of V5-$FZD_6$-mCherry surface crosslinking (CL) on the mobility of $FZD_6$-GFP in the presence of **a, b** pan-DVL siRNA ($P < 0.0001$; $t = 6.854$; df = 53; $N = 33$ ROIs before CL; $N = 22$ after CL from three independent experiments) and **c, d** overexpressed DVL2-MYC ($P < 0.0001$; $t = 6.243$; df = 77; $N = 36$ ROIs before CL; $N = 43$ after CL from three independent experiments). **e, f** dcFRAP experiments in combination with chemical surface crosslinking show that the $FZD_6$ dimer mutant is still capable of assembling in an inactive state with DVL2-GFP ($P < 0.0001$; $t = 5.173$; df = 74; $N = 15$ ROIs before CL; $N = 61$ after CL from three independent experiments), but not with $G\alpha_{i1}$-GFP ($P = 0.5646$; $t = 0.5801$; df = 48; $N = 29$ ROIs before CL; $N = 21$ after CL from three independent experiments). Error bars give s.e.m.; ***$P < 0.001$; ns non significant (two-tailed $t$-test). **g** V5-$FZD_6$-mCherry and $FZD_6$-GFP expressed in HEK293 cells showed distinct co-localization predominantly in the membrane. The $FZD_6$-R511C mutation (predominantly localized to lysosomes and unable to interact with heterotrimeric G proteins[16, 25]) forced the wt $FZD_6$ to be retained intracellularly suggestive of dimer formation. Co-expression of the $FZD_6$-R511C mutant together with the dimer mutant of $FZD_6$ did not result in co-localization. Due to the lack of dimerization, the $FZD_6$ dimer mutant showed surface expression in the presence of $FZD_6$-R511C. Size bar—10 μm

a ligand-independent manner (Fig. 6h). Taken together, the biochemical analysis of both $FZD_6$ wt and dimer mutant with regard to their capacity to induce P-ERK1/2 in the absence of agonist reveals the overactive nature of the $FZD_6$ dimer mutant compared to wt.

## Discussion

The existence and functional relevance of GPCR dimers or higher order oligomers is generally accepted despite difficulties to transfer mechanistic insight from overexpression studies using recombinant receptors to endogenously expressed receptors in cell systems and *in vivo*. Hitherto, agonist-dependent dissociation/re-association of GPCR dimers has not yet been reported and this paradigm provides deeper insight into the potential role of dimer dynamics in the process of FZD and potentially GPCR activation and signal initiation.

Depending on the specific GPCR class, dimers vary in their receptor complex formation, signaling, trafficking and pharmacology[32, 33]. In the case of Class A GPCRs, it is unclear whether dimers or higher oligomers have any physiological relevance[5, 6]. Conversely, Class C GPCRs constitutively dimerize via rather well-understood mechanisms and with obvious functional consequences for the receptors[34, 35]. Class FZD receptors ($FZD_{1-10}$, SMO) have also been shown to form higher order structures[14, 15, 23, 36]. In general, the functional importance of FZD dimerization/oligomerization remains obscure. For $FZD_4$, it was shown that dimerization occurs in the ER and that $FZD_4$ mutants associated with vitreoretinopathy can retain wt

$FZD_4$ in intracellular compartments[14]. It was also shown that $FZD_1$, $FZD_2$, $FZD_3$ dimerize and that dimerization of $FZD_3$ and $FZD_7$ affects WNT/β-catenin signaling in *Xenopus laevis*[14, 15]. So far, FZD dimers are perceived as being static and the nature of the dimer interfaces is unknown. Due to the well-documented interaction between CRD domains of FZDs[37], CRD dimerization has been proposed to form the structural basis of FZD interaction[15]. On the contrary, $FZD_1$ and $FZD_2$ were shown to form dimers independently of the CRD[14] and our data are consistent with a CRD-independent interface in the TM region of $FZD_6$. That being said, our results do not rule out the possibility for an involvement of the CRDs in supporting dimer formation. Throughout the GPCR families, various interaction interfaces have been identified in different receptor subtypes ranging from extracellular domains to basically all TM domains of the 7TM cores and the intracellular helix 8[5]. However, the involvement of TM1 and helix 8 in $FZD_6$ dimerization seems unlikely since the mutational analysis supports a TM4/5 interface and the C termini of either protomer point in opposing directions according to the homology model (Fig. 2a; Supplementary Fig. 2,3). The ability of $FZD_6$ to dimerize irrespective of DVL and heterotrimeric G proteins, underscores that dimerization is receptor-intrinsic. It should be emphasized that despite all data supporting the idea that a dimer is the smallest oligomeric unit of $FZD_6$, we cannot directly exclude the presence and dynamics of higher order receptor complexes. We base our conclusions about dimerization on the ability to define and disrupt the TM4/5 interface and on the rationale that $FZD_6$-D365A, R368A, Y369A does not continue

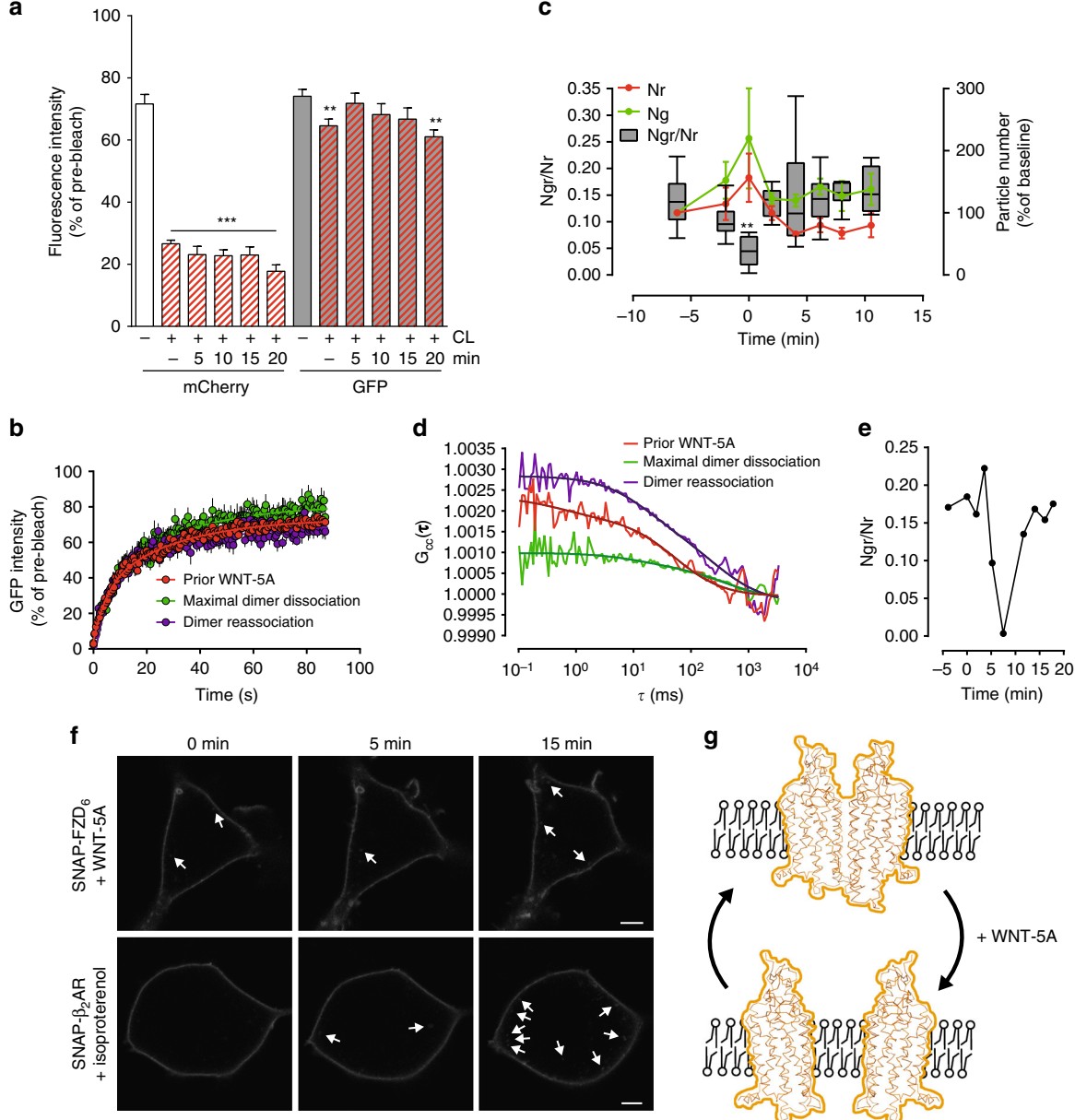

**Fig. 5** Agonist-dependent dynamics of FZD$_6$ dimerization. **a** dcFRAP in HEK293 cells expressing V5-FZD$_6$-mCherry and FZD$_6$-GFP allowed kinetic monitoring of receptor–receptor interactions in the absence of agonist and after 5, 10, 15, 20 min of WNT-5A (300 ng ml$^{-1}$) stimulation. $P = 0.0219$; F (5, 109) = 2.757. Before CL (20 ROIs); after CL (unstimulated = 38 ROIs, 5 min = 12 ROIs, 10 min = 18 ROIs, 15 min = 12 ROIs, 20 min = 15 ROIs). $N = 10$. Unstimulated (before and after CL) are from Fig. 1d and were performed in parallel with stimulation experiments. **b** Curves depict the recovery of GFP after CL comparing unstimulated (*red*), 5 min (*green*) and 20 min (*magenta*) after WNT-5A stimulation. **c** Dynamic FCCS experiments in HEK293 cells expressing V5-FZD$_6$-mCherry and FZD$_6$-GFP from 6 different cells are summarized by normalizing the agonist-induced minimum fraction of dimers from individual cells to 0 min ($N_{gr}/N_r$ represents ratio of FZD$_6$-GFP/V5-FZD$_6$-mCherry dimers (*green/red*, gr) over the total number of *red* molecules (*red*, r), representing the fraction of dimers present in the detection volume). Box plot presents data from the 25 to 75th percentile and whiskers provide min/max. $P = 0.0093$; F (7, 39) = 3.178. **$P < 0.01$ (one-way ANOVA, Fisher's LSD post hoc analysis). Total number of $N_r$ (*red*) and $N_g$ (*green*) particles in the detection volume over time. Values of $N_r$ or $N_g$ at the first time point were set to 100. **d** FCCS curves present cross-correlation of V5-FZD$_6$-mCherry and FZD$_6$-GFP from a representative cell prior to WNT-5A stimulation (*red*), at maximal dimer dissociation (*green*) and upon dimer re-association (*magenta*). When normalized to the *green* auto-correlation amplitude, the amplitudes of the three $G_{cc}$ curves revealed that 11, 6, and 21% of *red* molecules were also *green*, respectively. **e** Representative example of $N_{gr}/N_r$ after WNT-5A stimulation for a cell expressing wt FZD$_6$-GFP and wt FZD$_6$-mCherry. $t = 0$ refers to first measurement after the addition of WNT-5A. **f** HEK293 cells expressing SNAP-FZD$_6$ or SNAP-β$_2$-AR were pulsed with a cell-impermeable SNAP substrate to monitor internalization after WNT-5A (300 ng ml$^{-1}$, 15 min) or isoproterenol (10 μM). Arrows indicate internalized receptors. Scale bar = 5 μm. **g** Model of FZD$_6$ de- and re-dimerization upon WNT-5A stimulation. Error bar provides s.e.m

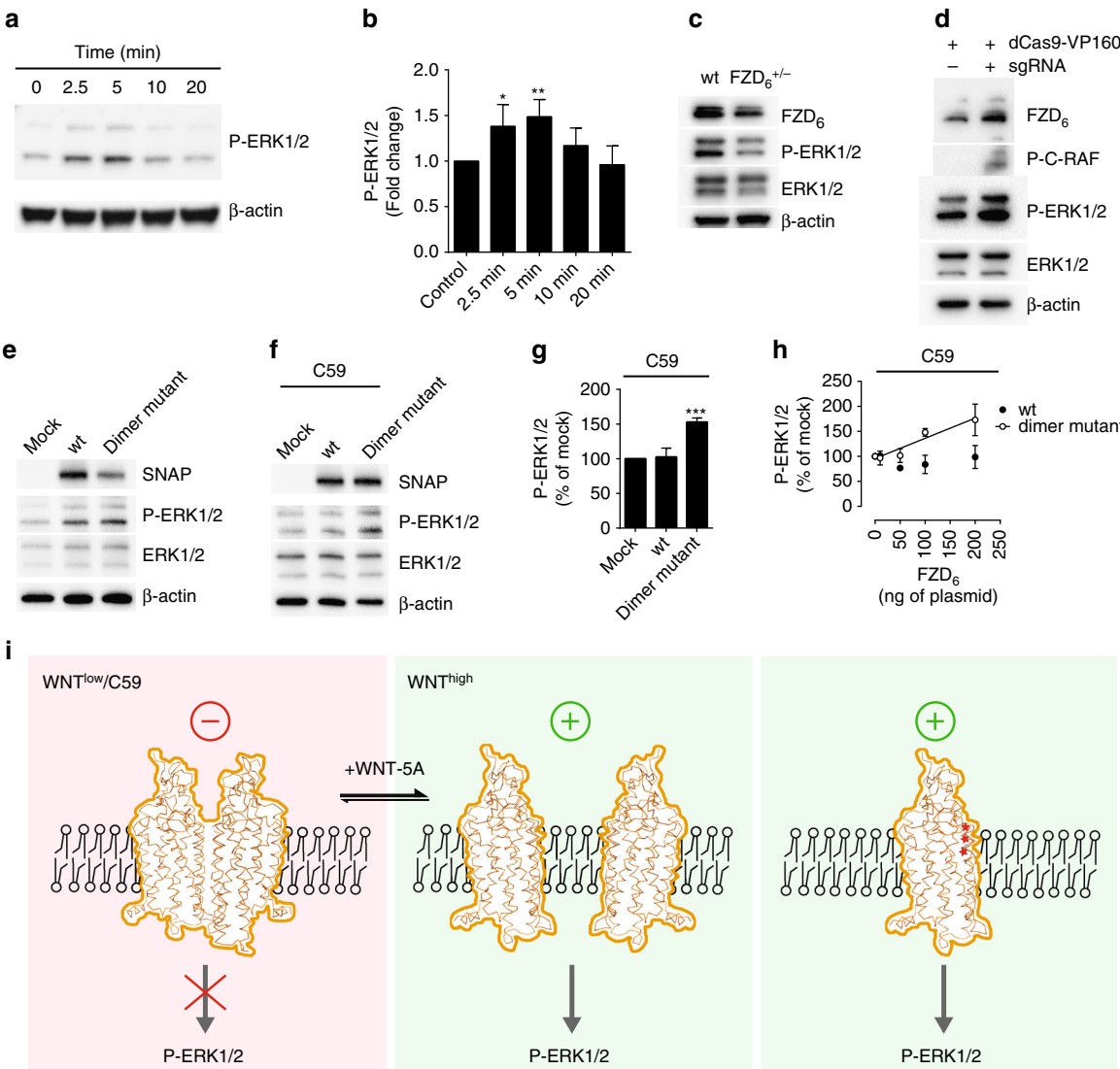

**Fig. 6** Relationship between $FZD_6$ dimerization and phosphorylation of ERK1/2. **a** Mouse lung epithelial (MLE-12) cells were stimulated with WNT-5A (300 ng ml$^{-1}$) for 0, 2.5, 5, 10, 20 min. Cell lysates were analyzed for P-ERK1/2 and β-actin by immunoblotting. Additional information in Supplementary Fig. 6. **b** Bar graph summarizes three independent experiments. Densitometry data were normalized to the unstimulated control. $P = 0.0037$; $F (4, 15) = 6.212$. $^*P < 0.01$; $^{**}P < 0.001$ (one-way ANOVA with Fisher's least significant difference (LSD) post hoc analysis). **c** Generation of a $FZD_6^{+/-}$ MLE-12 line using the CRISPR/Cas9 system resulted in decreased basal P-ERK1/2. **d** CRISPRa-mediated enhancement of $FZD_6$ expression led to increased P-C-RAF and P-ERK1/2. Additional information in Supplementary Fig. 6. SNAP-$FZD_6$ wt- and SNAP-$FZD_6$ dimer mutant-induced ERK1/2 phosphorylation in the absence **e** and presence **f** of the porcupine inhibitor C59 (5 μM, overnight) were probed by immunoblotting for the SNAP-tag, P-ERK1/2, total ERK1/2 and β-actin. **g** The bar graph summarizes densitometry data of P-ERK1/2 normalized to total ERK1/2 from 6 different experiments. $P = 0.0005$; $F (2, 15) = 13.08$. $^{***}P < 0.001$. Empty vector transfection was used as mock. **h** Increasing amounts of plasmid of either SNAP-$FZD_6$ wt or SNAP-$FZD_6$ dimer mutant in the presence of C59 show a dose-dependent increase in P-ERK1/2 only with the dimer mutant. Data from three independent experiments for the dimer mutant were fit to a linear least-squares model (slope = $0.40 \pm 0.10$; $R^2 = 0.5943$) and this line is shown superimposed. The ascending slope in the case of the dimer mutant, but not wt is significantly different from non-zero ($P = 0.0012$). Error bars give s.e.m. **i** Based on our experimental data, we propose a signaling paradigm where $FZD_6$ dimers confer the inactive receptor species, which can dynamically produce monomeric species showing a higher degree of activation, which feed into ERK1/2 signaling. This dissociation-dependent activation can be evoked by agonist (WNT$^{high}$) or mutation of the dimer interface (symbolized by *red stars*). WNT$^{low}$ conditions can be achieved in the presence of C59, which reduces $FZD_6$ wt signaling

to interact with $FZD_6$ wt in dcFRAP experiments—suggesting that other interaction modes enabling higher order complex formation do not exist. This conclusion is further corroborated by the analysis of the FCCS data, where the data fits with a model that assumes receptor dimerization.

The agonist-induced dynamic dissociation/re-association of the $FZD_6$ complex is clearly different from the rather stochastic, agonist-independent and transient interactions of Class A GPCRs measured by single-particle tracking in living cells[38–40]. Recent

data report antagonist-induced changes in the oligomeric organization of serotonin 5-HT$_{2C}$ receptors, where treatment with 5-HT$_{2C}$-selective antagonists reduced the complexity of higher oligomeric receptor assemblies assessed by spatial intensity distribution analysis[41]. Furthermore, agonist-induced rearrangements in the receptor-G protein complex composition of a calcitonin CTR receptor in relation to G protein association, coupling and activation support the idea of receptor dimer dynamics exemplified by a Class B GPCR[42]. Our findings extend

beyond the pre-existing concepts of constitutive, transient, or stable GPCR dimerization by introducing the idea that $FZD_6$ dimers dissociate and re-associate in a ligand- and receptor-activity-dependent manner.

In the current study, we map the dimer interface of $FZD_6$ to TM4/5 through mutational analysis, homology modeling and MD simulations. A similar interface has recently been described in the constitutive dimer of metabotropic glutamate receptor 2 (mGluR2), where agonist stimulation resulted in a structural rearrangement towards a smaller interface at TM6[35]. These findings were further corroborated in another FRET-based study investigating conformational dynamics of Class C GPCRs[43]. In the case of Class C receptors, protomer dissociation is prevented by covalent interaction at the N-terminus. When it comes to $FZD_6$, however, a structural rearrangement resembling that of mGluR2 results in dissociation because the receptor protomers are not covalently linked. The similarity between mGluR2 and $FZD_6$ dimer dynamics upon agonist stimulation supports the idea that dimer dynamics are integral to receptor activation and that this phenomenon might represent a more general principle than previously appreciated[35, 43].

Similar to our observations with $FZD_6$, the chemokine $CXCR_4$ receptor dimer structure[44] is also based on a TM4/5 interface localized to the upper regions of the helices with an interaction area of 850 $Å^2$. Although the molecular and structural details of Class FZD receptor activation are still unknown, extrapolation from the conformational changes observed for the $\beta_2$-adrenergic receptor bound to G protein or a nanobody surrogate suggests that activation could involve structural changes in TM5[45, 46], and thereby influence the homodimer interface of $FZD_6$. Interestingly, $D365^{5.41f}$, $R368^{5.45f}$, and $Y369^{5.45f}$ in $FZD_6$, which upon mutation influenced dimerization, are also in close proximity to the region in TM5 that undergoes a conformational change upon activation of the $\beta_2$-adrenergic receptor.

Present and previous dcFRAP experiments on $FZD_6$ dimer and G protein dissociation, respectively, argue that both processes follow similar kinetics[16]. In addition, we employ the TM1 and TM5 minigenes to elucidate dimer formation and the connection between dimerization and signaling (Supplementary Fig. 7). On the level of endogenous receptor expression in MLE-12 cells, we suggest that $FZD_6$ expression is required to maintain basal ERK1/2 phosphorylation—a pathway that is intrinsically linked to cell proliferation, survival, and repair[47]. In combination with the strong $FZD_6$ membrane expression in bronchial epithelium recently reported by the Human Protein Atlas project (http://www.proteinatlas.org/ENSG00000164930-FZD6/tissue/bronchus), these data argue that $FZD_6$ dimers could be important for the maintenance of a functional epithelium in the respiratory tract.

The detection, visualization, and analysis of GPCR dimers in cells endogenously expressing the receptors present a true challenge, which is not easily overcome. Employing both dcFRAP and FCCS with high and low expression of tagged receptors, we depict agonist-induced dimer dynamics in living cells. Translation of our findings to MLE-12 cells is possible using the validated dimer-interfering peptides. The TM5 minigene-mediated reduction in P-ERK1/2 in MLE-12 cells (Supplementary Fig. 7) indicates that peptide-induced $FZD_6$ dimer disruption is neither equivalent with agonist-induced dissociation nor is it sufficient to evoke signaling. One possible explanation is that the TM5 minigene disrupts the dimer through interaction with the $FZD_6$ TM4 and corrals the individual protomers in a negative allosteric modulatory manner to maintain the protomers in an inactive conformation (Supplementary Fig. 7). This hypothesis is further corroborated by the finding that the dimer mutant confers constitutive activity to the ERK1/2 pathway. Building on this idea, this could imply that the two protomers in a $FZD_6$ dimer maintain an inactive state in a bidirectional, negative allosteric fashion—a model that requires further experimental validation. Here, we conclude that the dimeric complex represents the inactive, ligand-responsive form of $FZD_6$ whereas the agonist-induced appearance of monomers or the increase of monomers by dimer-interfering mutations evokes downstream signaling (Fig. 6i). While wt $FZD_6$ forms an inactive-state complex with DVL and $G\alpha_{i1}/G\alpha_q$ proteins[16, 26], it is not possible to detect the assembly of the $FZD_6$-mCherry dimer mutant with $G\alpha_{i1}$-GFP by dcFRAP (Fig. 5f). Nevertheless, the $FZD_6$ dimer mutant still interacts with DVL (Fig. 5e) and signals to ERK1/2 in an agonist-independent manner when over-expressed in HEK293 cells (Fig. 6g). This apparent conundrum can be explained by the switch from an inactive-state receptor/G protein assembly to a collision-coupling mechanism in the predominantly monomeric dimer mutant as it would be expected from the ternary complex model[48]. In a complementary way, calcitonin GPCR (CTR) complex dynamics support our proposed model for $FZD_6$ because (i) the dimeric CTR appears to be the ligand-interpreting species and (ii) that increasing G protein interaction/coupling in the presence of agonist resulted in the loss of dimeric CTR[42].

Our data present a new kinetic perspective for the activation of Class FZD receptors where agonist stimulation leads to the dissociation and re-association of the receptor dimer. Here, we argue that the $FZD_6$–$FZD_6$ dimer is relevant in living cells and that constitutive signaling to ERK1/2 can be affected by modulating the dimeric status of $FZD_6$. Most importantly, we provide evidence that the monomeric rather than the dimeric form of $FZD_6$ is the active signal initiating unit of the receptor complex. This is in agreement with several lines of evidence supporting that the active conformation observed in GPCRs is associated with the monomeric receptor rather than a dimer[4, 42, 46, 49, 50].

FZDs present an attractive target for drug development, even though no small compounds targeting these receptors have been reported so far[8–10]. However, targeting the dimer interface to impede with receptor activation might offer novel therapeutic possibilities. Further studies will be required to fully understand the function of dynamic FZD dimerization for the agonist-induced initiation of different signaling avenues. Also, the phenomenon of dynamic and agonist-sensitive dimerization might not be restricted to Class F receptors, but could provide a more general concept of GPCR signal transduction.

## Methods

**Cell culture and transfections.** HEK293 and MLE-12 cells (ATCC) were cultured in DMEM supplemented with 10% FBS, 1% penicillin/streptomycin, 1% L-glutamine (all from Invitrogen Technologies) in a humidified $CO_2$ incubator at 37 °C. All cell culture plastics were from Sarstedt, unless otherwise specified. HEK293 cells were used because of their advantageous transfectability, their adherence and their appearance making them amenable to live cell imaging experiments. For live cell imaging experiments, cells were seeded on 35 mm ECM gel-coated (1:300, Sigma-Aldrich) glass bottom dishes (MatTec or Greiner Bio One 4 compartment 35 mm glass bottom dishes). Cells were transfected with Lipofectamine 2000 24 h prior to analysis. Treatment with the porcupine inhibitor C59 (2-[4-(2-Methylpyridin-4-yl)phenyl]-N-[4-(pyridin-3-yl)phenyl]acetamide) was performed at 5 μM overnight[24]. Treatment with pan-DVL siRNA was done as previously reported[16]. Absence of mycoplasma contamination is routinely confirmed by PCR using 5′-ggc gaa tgg gtg agt aac acg-3′ and 5′-cgg ata acg ctt gcg act atg-3′ primers detecting 16 S ribosomal RNA of mycoplasma in the media after 2–3 days of cell exposure.

$FZD_6$-GFP and $FZD_6$-R511C-GFP were from Niklas Dahl (Uppsala University, Uppsala, Sweden)[25]. GFP-tagged human $FZD_6$ and $FZD_6$-R511C were re-cloned into the pmCherry-N1 vector using the restriction enzymes BglII and AgeI. The following signal peptide (5′-cds: atg gtc ccg tgc acg ctg ctc ctg ctg ttg gca gcc gcc ctg gct ccg act cag acc cgg gcc-3′) was cloned into the pmCherry-N1 plasmid using HindIII and KpnI restriction sites. V5-$FZD_6$ was cloned using the following primers and inserted with KpnI and AgeI restriction sites into the aforementioned

pmCherry-N1 construct. Forward primer: 5′-acc ggt acc ggc aaa ccg att ccg aac ccg ctg ctg ggc ctg gat agc act cac agt ctc ttc acc tgt ga-3′. Reverse primer: 5′-gac cgg tgg agt atc tga atg aca acc acc t-3′. GFP-tagged human FZD$_6$ and FZD$_6$-R511C were re-cloned into the pmCherry-N1 vector using the restriction enzymes BglII and AgeI. The following signal peptide (5′-cds: atg gtc ccg tgc acg ctg ctc ctg ctg ttg gca gcc gcc ctg gct ccg act cag acc cgg gcc-3′) was cloned into the pmCherry-N1 plasmid using HindIII and KpnI restriction sites. V5-FZD$_6$ was cloned using the following primers and inserted with KpnI and AgeI restriction sites into the pmCherry-N1 construct. Forward primer: 5′-acc ggt acc ggc aaa ccg att ccg aac ccg ctg ctg ggc ctg gat agc act cac agt ctc ttc acc tgt ga-3′. Reverse primer: 5′-gac cgg tgg agt atc tga atg aca acc acc t-3′. SNAP-FZD$_6$ was a gift from M.M. Maurice (Utrecht University Medical Center, The Netherlands). SNAP-$\beta_2$-adrenergic receptor was from Davide Calebiro (University of Würzburg, Germany). CB$_1$R-GFP was from Zolt Lenkei (Ecole Supérieure de Physique et Chimie Industrielle–Paristech, Paris, France); DVL2-MYC was from S.A. Yanagawa (Kyoto University, Kyoto, Japan). G$\alpha_{i1}$-GFP was from Mark M Rasenick (University of Illinois, Chicago, USA). Plasmids encoding $\beta$ and $\gamma$ subunits were from www.cdna.org. The mCherry–TM–GFP stoichiometer was synthesized as a gBlock Gene Fragment by Integrated DNA Technologies (IDT, Coralville, Iowa, USA) and was composed of an NheI restriction site, a Kozak sequence, the Ig$\kappa$ signal sequence, a 6×His and a 3×Gly linker, an EcoRI restriction site, the coding sequence for mCherry, a BamHI restriction site, a 3×Gly linker, the platelet-derived growth factor receptor TM domain, a 3×Gly linker, an XbaI restriction site, the coding sequence for EGFP (humanized), an ApaI restriction site, a STOP codon, and a PmeI restriction site. The sequence was amplified by PCR using a forward primer: 5′-gct agc tga gct agc tcc acc atg-3′ and a reverse primer: 5′-aga agg cac agt cga ggc tga tca g-3′ and subcloned into pcDNA3.1 using NheI and PmeI restriction enzymes.

GeneArt Site-Directed Mutagenesis System (Life Technologies) was used for the insertion of all mutations listed in this publication. All constructs were confirmed by sequencing. The minigene sequences TM1: KKKSFIGTVSIFCLCATLFTFLTFLIKKK and TM5: KKLDASRYFVLLPLCLCVFVGLSLLLAGIISKKK were cloned into pcDNA3.1(+) (Invitrogen), including a signal sequence and an HA tag at the N-terminus. Lysines were added to support membrane embedding.

## FRAP and cellular imaging

For dual color FRAP experiments, cells grown in four compartment 35 mm glass bottom dishes were washed three times with BE buffer[19] (150 mM NaCl, 2.5 mM KCl, 10 mM HEPES, 12 mM glucose, 0.5 mM CaCl$_2$, and 0.5 mM MgCl$_2$). Then, the cells were incubated at RT for 30 min with biotin-conjugated goat polyclonal anti-V5 antibody (Bethyl) diluted 1:100 in BE buffer supplemented with 2.5% bovine serum albumin. For antibody-mediated cross-linking, avidin was added and incubated for 15 min and washed two times with BE buffer before the experiments were performed. In Fig. 4, dcFRAP experiments are presented that employ chemical surface crosslinking to visualize assembly of FZD$_6$ with DVL or heterotrimeric G proteins according to previously published protocols[16]. Live cell imaging experiments were carried out using a Zeiss 710 laser scanning microscope. Images were acquired using a 40×, 1.2 NA C-Apochromat objective and the 488 nm and the 561 nm laser lines were used to excite GFP and mCherry fluorophores, respectively. Photobleaching was performed by 100% laser illumination of a $1.80 \times 1.80$ μm area placed over the cell plasma membrane. Fluorescence was measured pre- and post-bleach using low-intensity illumination for a total time period of 100 s. Average pixel intensity was measured using the ZEN2013 software, corrected for background fluctuations and bleaching artifacts and normalized to pre-bleached intensity[51]. The recovered mobile fraction ($F_m$) was calculated as follows: $F_m = (I_P - I_0)/(I_I - I_0)$, where $I_I$ is the initial intensity measured prior to bleaching, $I_0$ is the immediate fluorescence intensity post-bleaching and $I_P$ is intensity value post-bleaching. The mobile fraction was defined by the average of the fluorescence recovery assessed between time 85–101 s (including pre-bleach measurements). We excluded experiments in which the extracellularly tagged receptors were not sufficiently immobilized after the cross-linking procedure (cutoff at 40% crosslinking-induced reduction in mobile fraction). Kinetic determination of mobile fractions of crosslinked V5-FZD$_6$-mCherry and FZD$_6$-GFP with WNT stimulation over time were performed in one cell culture well. Measurements for each time point were performed on separate cells.

SNAP-FZD$_6$ and SNAP-$\beta_2$-AR were imaged in living cells using a Zeiss LSM710 equipped with a 40×, 1.2 NA C-Apochromat objective. The cell-impermeable SNAP substrate SNAP-Surface 549 (New England Biolabs) was added (1:1000, 15 min), washed twice in BE buffer and imaged using the 561 nm laser line. SNAP-FZD$_6$ and SNAP-$\beta_2$-AR were stimulated with 300 ng ml$^{-1}$ WNT-5A and 10 μM isoproterenol, respectively, and fluorescence was monitored over time.

## Fluorescence correlation spectroscopy

FCS experiments were performed on an FCS/FCCS-equipped Zeiss 780 confocal microscope (Zeiss, Jena, Germany). In contrast to FCCS experiments, HEK cells expressing GFP-tagged constructs of either FZD$_6$ wt or FZD$_6$ dimer mutant were incubated with the porcupine inhibitor C59 (5 μM, overnight) in order to remove the presence of endogenous WNTs. Two measurement series were performed on each cell, each series consisting of 15 measurements of 4 s each. The auto-correlation curves were then fit to a model including one diffusion component in order to determine the transit diffusion time

($\tau_D$) and times shorter than 1 ms were excluded from the fit. Fitting was carried out using the Zen 2012 software (Zeiss, Jena, Germany).

## Fluorescence cross-correlation spectroscopy

FCCS measurements to detect FZD$_6$-GFP-V5-FZD$_6$-mCherry dimers (CB$_1$-GFP as control) in HEK293 cells (Fig. 1f–h) were performed on a Confocor 3 system (Zeiss, Jena, Germany). An Ar-ion laser (488 nm) and HeNe laser (543 nm) were focused through a C-apochromat 40×, NA 1.2 objective. The fluorescence was detected by two avalanche photodiodes after passage through a dichroic mirror (HFT 488/543/633), a pinhole (edge-to-edge distance 70 μm for FCCS) in the image plane, a beam splitter (NFT 545), and an emission filter in front of each detector (BP 505–530 IR and 560–610 IR). The excitation power before the objective was within the range of 1–10 μW for both the 488 nm-line and the 543 nm-line. The lateral radii $\omega_G$ and $\omega_G$ of the confocal volumes of the 488 nm laser and the 543 nm laser were fixed to 222 and 272 nm, respectively, determined by control experiments with Rho110 and TMR in double distilled water.

HEK293 cells were grown over-night in Lab-Tek eight-well chamber-glass bottom slides (Nunc, Thermo Scientific, Langensbold, Germany) coated with matrigel. Data were acquired in 10 s intervals. Collection intervals containing abnormal fluorescence peaks (presumably resulting from aggregates or ROIs where the overall fluorescence was decaying, putatively due to membrane movements) were discarded from the overall analysis. The total included measurements time ranged from 20 to 80 s per cell. Measurements were only undertaken on cells where the fluorescence was well-localized to the cell surface membrane, as judged by visual inspection in the wide-field and confocal mode. Autofluorescence, as well as fluorescence from both GFP and mCherry in the intracellular region was negligible. The number of fluorescent receptors assessed in each experiment varied from 10–1000 receptors per μm$^2$ in each HEK293 cell.

Dynamic FCCS measurements of WNT-dependent dissociation and re-association of FZD (Fig. 5c–e) were performed on an FCS/FCCS-equipped Zeiss 780 confocal microscope (Zeiss, Jena, Germany). The instrument details were the same as those of the Confocor 3 described above, with the exception that 561 nm excitation was used for excitation of mCherry, that GFP emission was detected from 500 to 550 nm, and that mCherry emission was detected from 600 to 690 nm. Experiments on cells were performed as described above, with each measurement point consisting of 3–6 measurements of 10 s each. In contrast to dcFRAP, FCCS experiments allowed assessment of receptor dynamics over time in single cells.

## FCCS analysis

Analysis of initial FCCS measurements confirming the presence of FZD$_6$-GFP/V5-FZD$_6$-mCherry dimers (Fig. 1h–j): In FCS/FCCS measurements, fluctuations in the detected fluorescence intensity $F(t)$ are typically generated as molecules diffuse in and out of a focused laser beam. These fluctuations, $\delta F(t)$, are autocorrelated according to $G(\tau) = \langle \delta F(t) \delta F(t+\tau) \rangle / \langle F(t)^2 \rangle$, or in case of two wavelength separated fluorescence intensities $F_G(t)$ and $F_R(t)$, cross-correlated according to $G_{GR}(\tau) = \langle \delta F_G(t) \delta F_R(t+\tau) \rangle / (F_G(t) F_R(t))$. In the case of two fluorescently labeled species, with concentration $c_G$ and $c_R$, and their interacting complex, with concentration $c_{GR}$, which are diffusing on a two-dimensional surface, the following equations hold[52]:

$$
\begin{cases}
G_{GR}(\tau) - 1 = \dfrac{\frac{A_G A_R}{A_{GR}} c_{gr} \times \text{Displ}(\tau) \times \text{Diff}_{gr}^{GR}(\tau) + A_G K \left( c_g x_g^2 \text{Diff}_g^G(\tau) + c_{gr} \text{Diff}_{gr}^G(\tau) \right)}{\left( A_G \left( x_g c_g + c_{gr} \right) + BG_G \right) \left( A_R \left( x_r c_r + c_{gr} \right) + A_G K \left( x_g c_g + c_{gr} \right) + BG_R \right)} \\[14pt]
G_R(\tau) - 1 = \dfrac{A_R \left( x_r^2 c_r \text{Diff}_r^R(\tau) + c_{gr} \text{Diff}_{gr}^R(\tau) \right) + 2\frac{A_G A_R}{A_{GR}} K c_{gr} \text{Displ}(\tau) \times \text{Diff}_{gr}^{GR}(\tau) + A_G K^2 \left( x_g^2 c_g \text{Diff}_g^G(\tau) + c_{gr} \text{Diff}_{gr}^G(\tau) \right)}{\left( A_R \left( x_r c_r + c_{gr} \right) + BG_R + A_G K \left( x_g c_g + c_{gr} \right) \right)} \\[14pt]
G_G(\tau) - 1 = \dfrac{A_G \left( x_g^2 c_g \text{Diff}_g^G(\tau) + c_{gr} \text{Diff}_{gr}^G(\tau) \right)}{\left( A_G \left( x_g c_g + c_{gr} \right) + BG_G \right)^2}
\end{cases}
\tag{1}
$$

$$
\begin{cases}
\text{Diff}_u^G(\tau) = \left( 1 + \dfrac{4 D_u \tau}{\omega_G^2} \right)^{-1} \\[10pt]
\text{Diff}_u^R(\tau) = \left( 1 + \dfrac{4 D_u \tau}{\omega_R^2} \right)^{-1} \\[10pt]
\text{Diff}_{gr}^{GR}(\tau) = \left( 1 + \dfrac{4 D_{gr} \tau}{(\omega_G^2 + \omega_R^2)/2} \right)^{-1}
\end{cases}
\tag{2}
$$

Here, $A_G = \pi \omega_G^2$, $A_R = \pi \omega_R^2$ and $A_{GR} = \pi(\omega_G^2 + \omega_R^2)/2$ are the effective detection areas, respectively, where $\omega_G$ and $\omega_R$ are the radii of the detected fluorescence brightness distribution of the two excitation lasers and $\text{Displ}(\tau) = e^{-2r_0^2/(\omega_G^2 + \omega_R^2 + 8 D_{GR} \tau)}$ is the displacement function related to a non-perfect overlap between the two excitation lasers. $D_u$ is the diffusion coefficient of species u. $K$ is the crosstalk parameter, defined as the brightness ratio of the *green* and *red* species at the center of each foci, when detected in the *red* channel. BG$_G$ and BG$_R$ are the background fluorescence in the *green* and the *red* channel, respectively, when both lasers are on. The terms $x_g$ and $x_r$ are the

absorption cross-section ratios of the g and the gr species and the r and the gr species, respectively. The expression in Eq. 1 was globally fitted by a nonlinear least-squares optimization routine developed in Matlab (Mathworks Inc, Natick, MA) to the three correlation curves ($G_G(\tau)$, $G_R(\tau)$ and $G_{GR}(\tau)$) recorded in each FCCS experiment. In the analysis, a two-step procedure was followed where a set of experimental parameters was first determined by control experiments (expression of one fluorophore only): (1). In control FCCS experiments (with no gr species), $\omega_G$ and $\omega_R$ were fixed to the corresponding values found in the solution measurements and $c_{gr}$ was fixed to zero. The rest of the parameters ($c_g$, $c_r$, $D_g$, $D_r$, $D_{gr}$, $r_0$, and $K$) were free to vary. The values $x_g$ and $x_r$ were fixed to either 1 or 2, reflected by whether FZD-GFP or CB$_1$-mCherry form homodimers or not, resulting in four different values of $K$ for each measurement. 2). In the dimerization measurements, $K$ was fixed to the median value determined from the presumed control in each of the four cases described above. The parameters $\omega_G$ and $\omega_R$ were fixed to the values found in the solution measurements and $r_0$ was restricted to a value of 100–150 nm that was determined previously[52]. The values $x_g$ and $x_r$ were fixed to the same number (either 1 for monomers or 2 for dimers) in the FZD$_6$-GFP/FZD$_6$-mCherry experiments and all four combinations of 1 and 2 in the FZD$_6$-GFP/CB$_1$-mCherry experiments. For the analysis, only measurements were included that complied with the following exclusion criteria when fitted to a single species normal diffusion auto-correlation curve: the measurement time range per data point/cell was 20–80 s (that means two to eight 10 s measurements); the diffusion time through the detector volume, $\tau_D$, was between 1–150 ms; $R^2$ in fitting between 0.9 and 1; counts per molecule (CPM) 0.2–2.5 kHz/molecule and the brightness ratio CPM $X_{GFP}$/ CPM $Y_{mCherry}$ between 0.5–2.5—where $X$ and $Y$ is FZD$_6$ and FZD$_6$, respectively. The analysis is based on six independent experiments in which 25/90 (FZD$_6$–FZD$_6$/FZD$_6$-CB$_1$) cells were assessed resulting in hundreds of individual 10 s measurements, which are summarized in the presentation of 10/25 cells in Fig. 1j.

Assuming that $M$ is FZD$_6$-GFP (green) and $N$ is FZD$_6$-mCherry (red) and that all FZD$_6$ form dimers, then the probability of finding a green–red FZD$_6$ dimer is $p_{gr} = \frac{2 \times M \times N}{(M+N)(M+N-1)} \approx \frac{2 \times M \times N}{(M+N)^2} = \frac{(2c_{gr}+c_{gr})(2c_{rr}+c_{gr})}{2(c_{gg}+c_{gr}+c_{rr})^2}$, where $c_{gg}$, $c_{gr}$, and $c_{rr}$ are the corresponding concentrations of the only green FZD$_6$ dimers, the red and green FZD$_6$ dimers and the only red FZD$_6$ dimers. Given a population of $M + N$ FZD$_6$ molecules, we define the probability to pick one green as $M/(M + N)$, followed by picking one red as $N/(M + N - 1)$ OR to pick one red as $N/(M + N)$ followed by picking one green as $M/(M + N - 1)$. The concentration of dimers consisting of a red and a green FZD$_6$ is then $p_{gr} \times (M + N)/(2)$ (there are $(M + N)/2$ pairs). The corresponding concentration of dimers consisting of green–red FZD$_6$ is then $c'_{gr} = p_{gr} \times (c_{gg} + c_{gr} + c_{rr}) = \frac{(2c_{gr}+c_{gr})(2c_{rr}+c_{gr})}{2(c_{gg}+c_{gr}+c_{rr})}$. From the experiments, we determine the concentrations for $c_{gg}$, $c_{gr}$ and $c_{rr}$ and compare the concentrations $c_{gr}$ with the $c'_{gr}$ for each cell to get an estimation of how close the model is to experimental data:

$$\frac{c_{gr}}{c'_{gr}} = 2c_{gr} \times \frac{(c_{gg} + c_{gr} + c_{rr})}{((2c_{gg} + c_{gr})(2c_{rr} + c_{gr}))}$$

Analysis of the dynamic FCCS measurements of WNT-dependent dissociation and re-association of FZD$_6$: Auto-correlation and cross-correlation curves were analyzed as described above and the amplitudes $G_g(0)-1$, $G_r(0)-1$, and $G_{cc}(0)-1$ from each of the more than 300 measurements of 10 s duration were estimated by fitting to theoretical models. The non-perfect overlap between the red and the green detection volumes was not taken into account since the dynamic FCCS measurements focused on the relative changes in fractional binding. The fraction of red units carrying also a green label, $N_{gr}/N_r$, was estimated from the ratio $(G_{cc}(0)-1)/(G_g(0)-1)$[53]. For FCS/FCCS measurements, about 20 % of the measurements were deselected due to membrane movements or low molecular brightness (≤0.2 CPM per second (cpm)). Also, cells where the intensity ratio between the channels $I_g/I_r > 1$ were deselected.

With the settings described above for detection of emission, the crosstalk from the GFP channel to the mCherry channel, defined as the intensity detected in the mCherry channel divided by the intensity detected in the GFP channel, $I_r/I_g$, for a cell expressing only FZD-GFP, was 0.051 or 5.1 %. The cross-correlation amplitude $G_{0,x}$ was therefore corrected for crosstalk before calculating the relative cross-correlation amplitude $G_{0,x}/G_{0,g}$[54].

**Molecular modeling and molecular dynamics simulations**. The structure of SMO crystallized in dimeric form (PDB code 4JKV)[55] was used as a template to model the FZD$_6$ homodimer. The sequence of FZD$_6$ (Uniprot ID O60353) was aligned to that of SMO with ClustalX2[56]. Termini residues corresponding to those not resolved in the SMO structure were omitted and the alignment was manually edited to ensure the proper alignment of conserved motifs among Class F GPCRs. In the final sequence alignment, FZD$_6$ and SMO shared a 30% sequence identity in the TM region. Fifteen homology models of the FZD$_6$ homodimer were generated with MODELLER 9.11[57] and were further analyzed by manual inspection. In order to evaluate the structural stability of the selected model, three independent MD simulations were prepared and carried out with GROMACS 4.6.7[58]. As described previously[59], the dimer was inserted into a pre-equilibrated hydrated 1-Palmitoyl-2-oleoylphosphatidylcholine bilayer. Data were then collected in 100 ns production

runs of unrestrained MD, resulting in more than 300 ns of simulation data. Simulations were conducted using the OPLSAA force field[60] for the protein with TIP3P waters[61], for which bond lengths and angles were constrained using the SETTLE algorithm[62], while the LINCS algorithm was used to constrain bond length in the protein and lipids[63]. The double-pairlist half-ε method[64] was used to enable the compatibility of Berger parameters[65, 66] for the lipids. In all the simulations, the system was simulated in a hexagonal prism-shaped simulation box using periodic boundary conditions. A time-step of 2 fs was used and neighbor lists were updated every 10 fs for non-bonded interactions within 12 Å whereas the Particle Mesh Ewald (PME) method[67] was used for the treatment of Coulombic interactions beyond the cutoff. All simulations were carried out at 310 K using the Nose–Hoover thermostat[68] and at constant pressure of 1 bar using the semiisotropic Parinello–Rahman barostat[69] with a coupling constant of 2 ps and an isothermal compressibility constant of $4.5 \times 10^{-5}$ bar$^{-1}$.

**Peptide synthesis**. FZD$_6$ TM1 (WKKKSFIGTVSIFCLCATLFTFLTFLIKKK), TM4 (KKKVWFHAVAWGTPGFLTVMLLAMNKKK), TM5 (KKLDAS-RYFVLLPLCLCVFVGLSLLLAGIISKKK) and the TM5 D365A,R368A,Y369A (WKKLAASAAFVLLPLCLCVFVGLSLLLAGIISKKK–mutated residues underlined) peptides were purchased as custom-made peptides from Pepscan Therapeutics (Lelystad, Netherlands) or synthesized using a Liberty blue microwave-assisted peptide synthesizer (CEM) using standard Fmoc chemistry. The additional tryptophan at the N-terminus of the TM5 mutant peptide was added to allow spectroscopic concentration determination due to a lack of tyrosines/tryptophans. The peptides were cleaved from the resin using a mixture of 92% TFA, 5% TDW, and 3% triisopropylsilane for 2.5 h at room temperature. Peptides were purified using a Merck-Hitachi HPLC using a reverse-phase Vydac C18 semi-preparative column. Peptide identity was confirmed using MALDI-TOF mass spectrometry and their purity was determined by analytical HPLC. Peptide concentration was determined using a UV spectrophotometer (Shimadzu Kyoto, Japan).

**Analytical size-exclusion chromatography**. Analytical SEC was performed on an ÄKTA Explorer with a Superdex 30 analytical column (GE Healthcare) equilibrated with TDW + 0.01% DDM. Peptides were eluted with a flow rate of 0.8 ml min$^{-1}$ at room temperature. Elution profile was monitored by UV absorbance at 215 nm. For the binding experiments, the peptides were incubated together in TDW + 0.01% DDM at room temperature for 2 h.

**Circular dichroism and isothermal titration calorimetry**. CD spectra were recorded using a J-810 spectropolarimeter (Jasco, Easton, MD, USA) in a 0.1 cm quartz cuvette. Far-UV CD spectra were collected over 190–260 nm. ITC measurements were carried out at 20 °C on an ITC (VP–ITC 200, MicroCal). 820 μM FZD$_6$ TM5 and 165 μM FZD$_6$ TM4 were dissolved in 0.5% DDM in water. 2 μl of TM5 peptide were injected into 260 μl TM4 during each titration. The ITC data were analyzed with the Origin 7.0 software.

**CRISPR-Cas9-mediated gene editing**. The human codon-optimized Cas9 nuclease expression plasmid was obtained from Addgene (41815). The gRNA_GFP-T1 plasmid was obtained from Addgene (41820) and used as a template for generating FZD$_6$-specific sgRNAs. The GFP targeting sequence was exchanged by inverse PCR followed by DpnI digestion and T4 ligation. Specifically, a common forward primer 5′-phospho-gtt tta gag cta gaa ata gca agt taa aat aag g-3′ was used in combination with a target specific reverse primer (i.e., reverse complement target sequence without the PAM for sgRNA1 5′-aag cac ctt aggact tcc gg -3′ was fused to the 5′ end of a common reverse oligo 5′-cgg tgt ttc gtc ctt tcc aca aga t-3′) to generate pU6-sgRNA1 by using AccuPrime Pfx DNA polymerase (Life Technologies) on the gRNA_GFP-T1 plasmid. MLE-12 cells were seeded out at a density of $1.25 \times 10^5$ cells onto 24-well plates. The next day, cells were transfected with Cas9 and pU6-sgRNA1 at equimolar ratios using Lipofectamine 2000 (Life Technologies). They were subsequently incubated for 48 h after transfection before dissociation and serial dilution in order to obtain a clonal cell line. Clones were screened by immunoblotting and confirmed by genomic DNA sequencing.

**CRISPR activation**. The plasmid encoding dCas9(D10A;H840A) fused with the VP160 activation domain (10 tandem copies of VP16) was obtained from Addgene (48226). Three sgRNAs were designed to target the promoter of FZD$_6$ between −400 and −50 bp upstream of the TSS: sgRNA1 (5′-tgt cgc ctc cgc tcg cac cgc gg-3′), sgRNA2 (5′-aga gca gat ccc cgc gcc taa gg-3′) and sgRNA3 (5′-agg gca ctt ccc caa cgg cct gg-3′). All sgRNAs were cloned by inverse PCR using the gRNA_GFP-T1 plasmid as a template. MLE-12 cells were seeded out at a density of $1.0 \times 10^5$ cells onto 24-well plates. dCas9VP160 (250 ng) was co-transfected with all 3 sgRNA constructs (100 ng each) using Lipofectamine 2000. Cells were incubated for 72 h before collecting for analysis. Successful activation was confirmed by both qPCR and immunoblotting.

**Immunoblotting.** MLE-12 or HEK293 cells (ATCC) were plated in a 24- or 48-well plate at a density of 100,000 or 50,000 cells/well, respectively. After 24 h, cells were transfected using Lipofectamine 2000 according to the manufacturer's instructions. Cells were stimulated (24 h post-transfection) with 300 ng ml$^{-1}$ recombinant carrier-free WNT-5A (CF WNT-5A; R&D Systems). Stimulation was stopped by the addition of lysis buffer (1% NP-40, 150 mM NaCl, 50 mM Tris/HCl pH 8). Lysates were sonicated and analyzed by 7.5 or 10% Mini-PROTEAN TGX precast polyacrylamide gels (Bio-Rad) and transferred to PVDF membranes using the Trans-Blot Turbo system (Bio-Rad). After blocking with 5% milk or BSA in TBS-T, membranes were incubated with primary antibodies in blocking buffer: mouse anti-β-actin (1:30,000; Sigma #A5441), rabbit anti-FZD$_6$ (1:1000; NovusBio #NBP1-00830), rabbit anti-Phospho-p44/42 MAPK (ERK1/2) (Thr202/Tyr204) (1:1000; Cell Signaling Technology #9101 L), rabbit anti-p44/42 MAPK (ERK1/2) (1:1000; Cell Signaling Technology #9102), rabbit anti-Phospho-C-RAF (Ser338) (1:1000; Cell Signaling Technology #56A6), rabbit anti-SNAP tag (1:1000, New England Biolabs #P9310S), and rabbit anti-HA (1:4000, abcam #ab9110) overnight at 4 °C. Proteins were detected with horseradish peroxidase-conjugated secondary antibodies (goat anti-rabbit and goat anti-mouse (Pierce)) and Clarity Western ECL Blotting Substrate (Bio-Rad). All uncropped immunoblots can be found in the Supplementary Information.

**Statistical analysis.** Statistical and graphical analysis was performed using Graph Pad Prism 5 software. Data were analyzed by two-tailed $t$-test or one-way ANOVA with Fisher's least significant difference post hoc analysis. Curve fitting of FRAP data was done with a two-phase association nonlinear function using the least square fit. Data for receptor titration in the presence of C59 and output to P-ERK1/2 in Fig. 6h was fit to a linear regression model. All experiments were repeated at least three times in independent experiments (not representing technical replicates). For single cell analysis (dcFRAP), the number of the individual ROIs obtained from several cells in at least three independent experiments is provided in the figure legends. Data from individual ROIs are summarized for data presentation providing the sum of independent observations. Significance levels are given as: *$P < 0.05$; **$P < 0.01$; ***$P < 0.001$. Data in FRAP curves and bar graphs (FRAP) are presented as mean ± s.e.m.

**Data availability.** The data that support the findings of this study are presented within the article and its Supplementary Information file and from the corresponding author upon reasonable request. All constructs originally described in this study can be obtained and used without limitations for non-commercial purposes on request from the corresponding author.

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

## Acknowledgements

Nevin Lambert is kindly acknowledged for his scientific input and help in creating the stoichiometer construct. We thank Marin M. Jukic for his input in statistical analysis. The work was supported by grants from Karolinska Institutet, the Science for Life Laboratory, the Swedish Research Council (2007-2769, 2007-5595, 2011-2435, 2013-5708, 2015-02899), the Swedish Cancer Society (CAN2011/690, CAN2014/659), the Knut & Alice Wallenberg Foundation (KAW2008.0149; 2008.0139), the Swedish Foundation for Strategic Research (ICA10-0098), the Board of Doctoral Education at Karolinska Institutet (JP), Engkvist's Foundations, Foundation Lars Hiertas Minne, Swedish Royal Academy of Sciences/Foundation Hierta-Retzius Fond, the Czech Science Foundation (13-32990S), the Program "KI-MU" (CZ.1.07/2.3.00/20.0180) co-financed from European Social Fund and the state budget of the Czech Republic and the Marie Curie ITN WntsApp (Grant no.608180; www.wntsapp.eu). D.R. is funded by a postdoctoral fellowship from the Sven och Lilly Lawski Foundation (N2014-0049). Computational resources were provided by the Swedish National Infrastructure for Computing (SNIC) and National Supercomputer Centre (NSC) in Linköping. J.C. and D.R. participate in the European COST Action CM1207 (GLISTEN). A.F. is supported by a grant from the Israel Science Foundation.

## Author contributions

J.P., S.C.W., D.R., P.M., J.C., S.W., J.S., N.L., and A.V. performed experiments, produced data for presentation. J.P., S.C.W., D.R., P.M., J.C., S.W., N.L., A.V., A.F., and G.S. planned experiments, analyzed data, and compiled figures. S.C.W., J.P., and G.S. wrote the paper.

## Additional information

**Competing interests:** The authors declare no competing financial interests.

