## [Peer Review File · Nature Communications]

Editorial Note: Parts of this peer review file have been redacted as indicated to remove third-party material where no permission to publish could be obtained.

Reviewers' comments:

Reviewer #1 (Remarks to the Author):

Review Nat.Comm. 2016

The manuscript of Peteren et al. entitled: „Agonist-induced dimer dissociation as a macromolecular step in G protein-coupled receptor signaling“ represents an interesting study using highly sophisticated biophysical assays to study dimerization of Frizzled 6 (FZD6) leading to a novel concept that Agonist binding induces dimer dissociation of Frizzled receptors which leads then to downstream signaling.

A: The long standing question of the signaling function of the dimerization of GPCRs may get a novel answer for Frizzled receptors, which mediate WNT signaling. Gunnar Schulte and his team measured FZD6 dimerization by both Fluorescence Correlation Spectroscopy and mobility restriction assays by means of dual color fluorescence recovery after photobleaching (dcFRAP). Using these assays the authors identified receptor mutants that were impaired in respect to their ability to interact with each other. The mutagenesis was guided by structural models of FZD6 dimers, which were based on the dimer orientation seen in the crystal structure of Smoothed. The authors concluded that the dimer interface was centered around TM4 and TM5. Peptides corresponding to TM5, but not those corresponding to TM1 interfered with receptor dimerization. Control experiments demonstrating proper membrane insertion of these peptides together with isocalometric data showing interaction of TM5 peptides with receptors convincingly demonstrate that these peptides were specifically interacting with receptors. Moreover, the authors show experimental evidence, that WNT5a mediated activation of FZD6 resulted in a transient attenuation of receptor dimerization. Finally, the authors claim basal ERK1 phosphorylation was increased in FZD6 dimerization deficient mutants, whereas over expression of TM5 reduced ERK1-phosphorylation. Based on these data the authors propose that FZD6 dimer dissociation either by WNT5a or introduction of interface mutants will induce downstream signaling.

B. This is a very appealing novel model of FZD receptor function. Even though the presented experimental evidence is quite convincing, the idea and concept is of high novelty and originality.

C, D Also the methods used are highly sophisticated and very useful to address the topics of the study. As pointed out below there are a few open issues with the analysis and interpretation of the data.

E. The obtained results are highly interesting, however the authors need to address some issues in respect to validate their model and interpretations. Some important issues that need further clarification:

F:

1. Throughout the manuscript the analysis of the dcFRAP seems to be suboptimal:

a: Why do the authors analyze just the recovery at 55-65 s instead of analyzing kinetics. The reasoning for this comment is that the analysis of kinetics would give some information on the stability of the dimer complex. The FRAP curves show that at 55-65 s only a small fraction of green receptors is engaged in long-lived receptor dimer formation. If you look just on this late time point, one could argue that the majority of receptors exists as monomers or unstable dimers.

b. Based on the finding that the immobilization of redFZD6 by crosslinking causes a significant reduction in FRAP of green FZD6 receptors and that in similar experiments using green CB1-R no significant reduction of the green FRAP curve was detected the authors claim that their data demonstrate dimerization of FZD6. To my understanding they must demonstrate a significant difference between control (CB1-R-GFP) and FZD6-GFP. Even though statistics will be more difficult, looking on Suppl. Fig 1 b (which by the way should be included in Fig.1 because it is the proper control) it is obvious that we cannot conclude that AB-crosslinking doesn't affect CB1-R mobility at all. Throughout the manuscript it would be important to directly compare controls with corresponding conditions. Also it is not indicated which statistical method was used for which Graph.

2. please show FRAP curves that underlie data from Fig 4a, at least those for the time points 0 and 5 min (GFP only would be sufficient). The reader wants to see what is the effect of agonist on

the receptor interactions.

3. Why are the FCCS blots in Fig 1 g and 4 b,c are blotted in a different way?

4. The negative effect of TM5 expression on P-ERK1/2 (Fig.6e,f) is inconclusive. If the effect would have been the opposite, the authors would have also claimed that it would support their claim. I would rather prefer to show it in the supplements.

5. Any explanation why the effect on crosslinking on FZD-GFP FRAP curves seem to be larger in Fig. 5 compared to Fig. 1?

6. Last but not least important: The authors did not show whether the dimerization deficient mutant is less or more active in the presence of WNT5a. This is very critical in respect to interpretation of the importance of dimerization for the signaling phenotyp of the mutant. Since there are numerous mutations in GPCRs that lead to an increase in basal activity (often accompanied by higher agonist affinity) it is important to know whether the mutation would lead to an increase in basal activity at the cost of the amplitude of the agonist induced activity.

Minor comments:

There are a couple of mistakes in the manuscript. Please reedit it for proper spelling: i.e.

Figure 5 legends: 1.8 asbling (assembling)

Results page 10: 210-214: asbly → assembly further , the % numbers are not explained: At first I thought the authors measure 63% assembly (dimer formation), but that is not what they mean.

„did not exhibit a reduction in the fluorescence recovery of the CB1-GFP mobile fraction upon crosslinking „

Line 223 „receptor returned to values rebliing pre-stimulation" (page 10)

G: The references give appropriate credit to previous work

H: The abstract /summary is well written and clear. Again the claims set out here need some validation as mentioned above

Reviewer #2 (Remarks to the Author):

This manuscript presents an interesting set of data about interactions of Frizzled6 in the membrane and apparent changes in response to the agonist WNT5a. The overall interpretation of the data is that the receptor is normally an inactive dimer that binds 1 DVL and 1 G protein and that this dimer separates to produce the active species that activates G protein and leads to ERK phosphorylation. On the one hand the story hangs together and is potentially an intriguing advance. On the other hand, most of the conclusions are built on data that show extremely small and in the case of the FCCS, variable and inadequately powered experiments. The language in the paper is problematic, with grammatical and syntax errors, neologisms or repeated typographical errors like "asbly" and "rebling" that occur throughout. The language problems bleed, at times, into an overexaggeration of the results, without the presentation of alternative explanations or caveats, or in some cases necessary controls.

A fairly trivial but early example of the overreach is the sentence on page 5: "Since simultaneous interaction of DVL and G proteins with one FZD6 molecule appears competitive, we propose a dimeric FZD quaternary structure, in which one protomer interacts with DVL and the other one

with the heterotrimeric G protein." This is fine to propose, but it is purportedly based on data about "one" molecule of FZD6 - how do the authors know what happens to single molecules of FZD6? These are conclusions drawn from ensemble FRAP data with very small changes in recovery times, done with what appears to be minimal attention to stoichiometry. I am fine with a hypothesis drawn from the data but not to argue it is based on knowing something about a single molecule of FZD6 is careless language that can mislead. Taken together these problems in language and argument lead to a sense of overreach that pervades the manuscript in which very specific molecular models are presented as established, when in fact they are an interpretation of the data, which are in many cases not sufficiently explored.

First the FRAP data. The authors achieve efficient immobilization of the target protomer with their crosslinking method, down to about 25% of control. The associated change in the second protomer is vastly smaller, from 74% to 64%, a 10% change instead of a 75% change. The nonsignificant CB1 control in SF1a shows a change of perhaps 5%. The differences in Figure 2 between what is significant and not are tiny and this is very worrisome - 2i shows a dimer control that is reportedly still a dimer whereas 2k shows that TM5 disrupts dimerization - yet the results seem essentially the same in value although statistically consistent with the interpretation. My interpretation, rather than a stable dimer interface is that this at best suggests a minor transient interaction that might slow the partner but in no way truly immobilizes it. (Compare with the much greater impact on the interaction with DVL shown in the SI.) If the dimer is stable then increasing the ratio of protomer A to protomer B should cause more and more of B to be associated with A and should enhance the effect. The only attention given to stoichiometry is the statement that equal amounts of receptor were necessary to see this, which doesn't seem consistent with the interpretation. Both protomers are theoretically the same except for their tags, so the more the A in the membrane, the more B should immobilize, and the more the B less B should immobilize. These controls seem to me to be essential. Another control would be to force interaction between the two protomers by chemical crosslinking or by the rapamycin system and see what happens. Efficient crosslinking should mimic a stable interface and lead to a much bigger effect of immobilizing A.

The FCCS data appear to be underpowered and problematic. The cell numbers are not sufficient for the analysis, as can be judged by the enormous spread of the data in Figure 1g and the quite substantial cross correlation of proteins that based on FRAP are said to not interact. This is not consistent with the methods (line 580) stating that cross talk is not an issue. The authors don't explain the y-axis in Fig 1g, which I presume it is a percent (i.e. they multiply the ratio in the rest of the FCCS y axes by 100), but it's not consistent with Fig 2, so this is difficult to interpret. The example in Fig 1h looks like the baseline is floating and the model function fit wouldn't be very good. The model function fits should be shown throughout. The cross-correlation amplitudes in Fig 2h aren't helpful without the autocorrelation functions. We need to see how they compare to their respective autocorrelation functions, not just the absolute comparison. In Fig 2g, the N is 6 cells, which is not enough based on the variation shown. The data for WT diverge greatly from Fig 1. Although the y axes problem noted above precludes certainty about this. The comparison in 4d is likely more legitimate because the same cell should have the same population over time, although, this should still be shown. The time-dependence is quite intriguing and consistent with the interpretation of the FRAP data, but we still need to see the N_r and N_g values over time to know if receptors are being internalized or photobleached during the long time measurement.

The methods contain multiple grammatical errors. The authors call a GFP-only experiment a 'negative control', but don't show any of the data. Line 574 - I have no idea what the authors mean by "were estimated by visual inspection." If taken literally, this is problematic.

Conclusions were drawn in a previous manuscript about the mutant 511C not interacting with G proteins based on FRAP data. In this manuscript we learn that 515C is not expressed at the plasma membrane but rather in lysosomes and other intracellular organelles. Thus arguing mechanistically from these data about G protein coupling is dangerous. The authors write: "Since neither R511C nor the dimer mutant of FZD6 were able to assemble with heterotrimeric G proteins, we collectively conclude that neither DVL nor G protein binding are required for the formation of an

inactive-state FZD6 dimer (Fig. 5g)." ("We collectively" means all the authors concluded on this together...) First we have the problem with 511C mentioned above. Second, a major finding of the paper is that the mutant that apparently cannot assemble into dimers still functions, in fact perhaps better than WT, to activate ERK, and since this effector pathway is reported to be partially sensitive to PTX, then the mutant does assemble with G protein, just not in a way that can be measured by the FRAP experiments. Statements like this are made like logical truths but when I try to break them down and understand what they mean in detail, problems like this arise.

The lack of significant movement at the interface of the molecular dynamics experiments is consistent with the interface being a reasonable one, but simulations of aggregate 100 ns are inadequate to speak to the stability of the interface. The length of simulation is not sufficient for sufficient evolution of a protein-protein interface, even if one wanted it to move. Thus, whether the interaction at the interface is transient or stable simply cannot be based on these data, which are again overinterpreted in my opinion as the general sense is that the authors propose a stable dimer until agonist comes, except when they want to increase the monomer concentration (see below) when they talk about an equilibrium suggesting that it is a transient interface that exchanges.

"Interestingly, the kinetics of WNT-5A-induced ERK1/2 phosphorylation matched the time course of FZD6 de- and re-dimerization, suggesting a role for dimerization in signaling." Suggesting a "possible" role would be less of a reach. A similar time course does not a mechanistic connection make.

"...confirming loss-of-function with gain of-function data (Fig. 6d)." I know what the authors mean but the sentence is problematic.

"Furthermore, overexpression of FZD6 WT results in quantitatively more monomeric receptors in a dynamic dimer/monomer equilibrium leading to enhanced signaling to ERK1/2. Overexpression of the FZD6 dimer mutant amplifies the number of monomeric receptors with stronger signaling output as a consequence." Increasing expression of WT is very difficult to interpret. It could make more monomers in to a transient dimer model - although for most of the paper it sounds like the model is for a very stable interaction until agonist. But of course overexpression of WT should also increase dimers, so it is also consistent with true dimers signaling but the inability of a WT protomer bound to a TM5 minigene product to signal. Many interpretations are possible but the statements are broad and taken as clear evidence of the proposed solution, without consideration of the caveats or alternatives. The dimer data are interesting and do suggest that this construct can signal. It is unclear to me from the data whether it is sensitive to agonist or not or just constitutively active, so again multiple interpretations are possible and perhaps agonist response needs a dimer. When one really gets down to the details, the conclusions become less clear, even though the data are interesting. More careful examination of the alternatives would drive experiments that might allow these alternatives to be differentiated rather than stated.

More examples of complicated and misleading sentences follow:

"Moreover, the FZD6 dimer can form independently of DVL or heterotrimeric G proteins (Fig. 5)." As noted above dead intracellular dimers or aggregates can form without the ability to show G protein interaction by FRAP and clearly the G protein interaction is missed with the dimer deficient mutant that signals, so this entire line of argument seems problematic and unnecessary.

"It is likely, but so far speculative, that agonist-induced dedimerization and G protein dissociation occur simultaneously, whereas the FZD6-DVL interaction could be static." Why is this likely and not just speculative? It requires data or a logical flow. As noted above, it isn't even clear to me that the monomer is the agonist responsive species. How do we know if internalization or some other change in response to agonist is giving a confound - as noted above the FCCS data is

wanting controls here as well to show that the autocorrelation is not altered.

"This is in agreement with several lines of evidence supporting that monomeric rather than dimeric GPCRs present the true active conformation (Fig. 6)4,41,44." The one true conformation is not a helpful concept. These papers show or infer that monomers CAN signal. They say nothing about what happens in the membrane and being monomeric isn't really a conformation anyway.

"Also, the phenomenon of dynamic and agonist-sensitive dimerization might not be restricted to Class F receptors, but could provide a more general concept of GPCR signal transduction." This comes after discussion about the differences and how Family A are all transient and not shown to respond to agonist (except this is contradicted by the discussion on alpha 2C) so unclear what is being said. The area of Family A dimerization of GPCRs is controversial and hotly debated and general statements like this are not helpful.

There was evidence previously for dimerization of other FZDs and that this dimerization is essential to get to the surface. For FZD6 the current data suggest that this is not the case as the dimer deficient mutant gets to the surface and signals, at least constitutively (see above on the need to assess its response to agonist). Thus, there is likely to be a diversity of mechanisms even for different FZDs. What the field needs is clear, logical, methodologically sound steps towards consensus. The authors don't discuss this difference of the FZD literature with their current inferences about FZD6 but instead make grand sweeping and largely uninterpretable claims about a general concept of GPCR signal transduction.

RESPONSE:

First of all, we would like to thank both reviewers for their qualified criticism. We have substantially revised the manuscript addressing most, if not all, comments raised. Before addressing the reviewers' criticism point by point, we would like to summarize the **major changes** that have been introduced to the manuscript and the study:

1. In order to improve the dcFRAP analysis, we have created a "stoichiometer" – an artificial protein carrying an extracellular mCherry, a single spanning transmembrane domain and an intracellular GFP. A similar approach to quantify the dcFRAP experiments was used in the following papers by Nevin Lambert (Qin et al., 2011; Qin et al., 2008):

Qin K, Sethi PR, Lambert NA. Abundance and stability of complexes containing inactive G protein-coupled receptors and G proteins. *FASEB J.* 2008 Aug;22(8):2920-7. doi: 10.1096/fj.08-105775.

Qin K, Dong C, Wu G, Lambert NA. Inactive-state preassembly of G(q)-coupled receptors and G(q) heterotrimers. *Nat Chem Biol.* 2011 Aug 28;7(10):740-7. doi: 10.1038/nchembio.642.

This stoichiometer construct allowed us to define the relative expression levels of the V5-FZD₆-mCherry over the FZD₆-GFP. Thereby, we could verify that only data are included in the analysis of dcFRAP experiments where the immobilized V5-FZD₆-mCherry was in excess of FZD₆-GFP. See also the new Supplementary Figure 1.

In the left graph provided below, all ROIs are plotted that were obtained from the dcFRAP experiments. The blue line presents data obtained with the stoichiometer providing a 1:1 ratio of mCherry and GFP. Only data below this line were included in our data analysis (see ROIs provided in the right graph, which is also included in Supplementary Figure 1a of the revised manuscript).

Average expression levels of mCherry : GFP in the different experimental conditions:

FZD₆-mCherry : FZD₆-GFP – 1.58 mCherry : 1 GFP

FZD₆-mCherry : FZD₆ dimer mutant-GFP – 2.03 mCherry : 1 GFP

FZD₆-mCherry : CB₁-GFP – 1.94 mCherry : 1 GFP

By exclusion of the data with a ratio of GFP>mCherry, we can be certain that we did not include any data points where an interaction would be missed if it existed.

2. The question of ligand dependency of the FZD₆-mediated ERK1/2 phosphorylation is highly interesting especially with regard to possible constitutive activity of the dimer mutant. Due to the lack of isoform selective FZD agonists/antagonists, the only ligands available are the

recombinant WNT proteins. However, these would – in addition to the overexpressed receptors – also activate the endogenously expressed FZDs rendering it impossible to distinguish responses mediated by the different receptor pools. In order to address if the wt-FZD₆, but not the FZD₆ dimer mutant, could be ligand dependent, we used the porcupine inhibitor C59 that results in the blockade of endogenous WNT secretion. In this way, we could create conditions that are WNT low and WNT high. In this experimental set-up, we were able to dissect the wt vs mutant-induced ERK1/2 phosphorylation (Fig. 6f, g) supporting the hypothesis that the dimer mutant exhibits constitutive activity. Unfortunately, we cannot at the present time determine whether the dimer mutant is still able to respond to WNTs. We can neither perform WNT-ligand binding assays (due to their high lipophilicity) nor are pharmacological tools available targeting Class Frizzled receptors. This raises the question of whether the dimer in addition to inhibiting the signaling output to ERK1/2 also plays a role in ligand recognition – a question that we currently cannot address.

3. An addition to the revised manuscript is the analysis of diffusion transit time τ_D by fluorescence correlation spectroscopy employing wt-FZD₆-GFP or dimer mutant FZD₆-GFP (Fig. 2i). This offers a clearer methodology to assess receptor complex size in the plasma membrane of living cells avoiding various different receptor populations (mixed wt/dimer mutant) as well as varying expression levels of mCherry- or GFP-tagged receptors.

Reviewers' comments:

Reviewer #1 (Remarks to the Author):

Review Nat.Comm. 2016

The manuscript of Peteren et al. entitled: „Agonist-induced dimer dissociation as a macromolecular step in G protein-coupled receptor signaling" represents an interesting study using highly sophisticated biophysical assays to study dimerization of Frizzled 6 (FZD6) leading to a novel concept that Agonist binding induces dimer dissociation of Frizzled receptors which leads then to downstream signaling.

A: The long standing question of the signaling function of the dimerization of GPCRs may get a novel answer for Frizzled receptors, which mediate WNT signaling. Gunnar Schulte and his team measured FZD6 dimerization by both Fluorescence Correlation Spectroscopy and mobility restriction assays by means of dual color fluorescence recovery after photobleaching (dcFRAP). Using these assays the authors identified receptor mutants that were impaired in respect to their ability to interact with each other. The mutagenesis was guided by structural models of FZD6 dimers, which were based on the dimer orientation seen in the crystal structure of Smoothed, The authors concluded that the dimer interface was centered around TM4 and TM5. Peptides corresponding to TM5, but not those corresponding to TM1 interfered with receptor dimerization. Control experiments demonstrating proper membrane insertion of these peptides together with isothermal titration calorimetry data showing interaction of TM5 peptides with receptors convincingly demonstrate that these peptides were specifically interacting with receptors. Moreover, the authors show experimental evidence, that WNT5a mediated activation of FZD6 resulted in a transient attenuation of receptor dimerization. Finally, the authors claim basal ERK1 phosphorylation was increased in FZD6 dimerization deficient mutants, whereas over expression of TM5 reduced ERK1-phosphorylation. Based on these data the authors propose that FZD6 dimer dissociation either by WNT5a or introduction of interface mutants will induce downstream signaling.

B. This is a very appealing novel model of FZD receptor function. Even though the presented experimental evidence is quite convincing, the idea and concept is of high novelty and originality.

RESPONSE: We very much value the appreciation of the model and novelty of our study and hope that the additions and improvements have addressed most of the very relevant criticism.

C, D Also the methods used are highly sophisticated and very useful to address the topics of the study. As pointed out below there are a few open issues with the analysis and interpretation of the data.

E. The obtained results are highly interesting, however the authors need to address some issues in respect to validate their model and interpretations. Some important issues that need further clarification:

F:

1. Throughout the manuscript the analysis of the dcFRAP seems to be suboptimal:

a: REVIEWER STATED: Why do the authors analyze just the recovery at 55-65 s instead of analyzing kinetics. The reasoning for this comment is that the analysis of kinetics would give some information on the stability of the dimer complex. The FRAP curves show that at 55-65 s only a small fraction of green receptors is engaged in long-lived receptor dimer formation. If you look just on this late time point, one could argue that the majority of receptors exists as monomers or unstable dimers.

RESPONSE: This point is of special interest to us since we originally hoped to be able to extract information about protein kinetics from the shape of the FRAP curves, which turned out to be much more complex than expected. The reasons for that are most likely the different behavior of molecules in free solution, where diffusion characteristics are well defined compared to the behavior of proteins diffusing in a 2D lipid environment. Too many factors affect protein mobility to draw conclusions quantitatively. However, the FRAP analysis allows us to define the mobile fraction of the protein of interest. We have especially taken the argumentation presented in the review Sprague and McNally 2005 (Trends in Cell Biology) (Sprague and McNally, 2005) into account, distinguishing between the early and the late phase in FRAP curves. The early phase (ascending intensity values during recovery) represents mostly the process of protein diffusion compared to the later phase (plateauing values in fluorescence recovery) that represents the binding between proteins of interest. In order to ensure that our data indeed represent the analysis of the mobile fraction and the binding between the FZD₆ species, we have changed our analysis to the time frame 85-101s (including 15 s pre-bleach measurements).

Also, with regard to the amplitude of the response in the dcFRAP assay, it is important to realize that the theoretical maximal possible response is at most 50% of the change in mobile fraction of the immobilized receptor. Of the four FZD₆ dimer species (red/red, green/green, red/green and green/red) only 50% can be assessed by this methodology (the measurable dimers need to be immobilized and detectable by GFP fluorescence). This 50% decrease in mobile fraction would represent stable dimers. For example, at 75% immobilization of the red fraction, at most 37.5% of the green can be affected. The values that we report are about a third of the fraction maximally expected for stable dimers. A further factor affecting the amplitude of the decrease in mobile fraction upon crosslinking is obviously the relative expression level of red and immobilized species over the green. This aspect has been addressed by introducing the stoichiometer as mentioned above – see also comment to reviewer #2.

b. REVIEWER STATED: Based on the finding that the immobilization of redFZD6 by crosslinking causes a significant reduction in FRAP of green FZD6 receptors and that in similar experiments using green CB1-R no significant reduction of the green FRAP curve was detected the authors claim that their data demonstrate dimerization of FZD6. To my understanding they must demonstrate a significant difference between control (CB1-R-GFP) and FZD6-GFP. Even though statistics will be more difficult, looking on Suppl. Fig 1 b (which by the ways should be included in Fig.1 because it is the proper control) it is obvious that we cannot conclude that AB-crosslinking doesn't affect CB1-R mobility at all. Throughout the manuscript it would be important to directly compare controls with corresponding conditions. Also it is not indicated which statistical method was used for which Graph.

RESPONSE: The figure with CB₁ control experiments has been included in Figure 1e, f, g. Furthermore, the curve shape of the CB₁ dcFRAP experiment changed upon data exclusion according to the mCherry-TM-GFP stoichiometer (see comments above). Given the fact that there was a greater excess of mCherry in CB₁ experiments compared to FZD₆ wt experiments where we see an interaction, the relative Δ revealed that there is really no interaction between FZD₆ and CB₁.

CB₁ experiment before CL $78,48 \pm 2,373\%$, n=29

CB₁ experiment after CL $77,12 \pm 2,601\%$, n=34

FZD₆ wt experiment before CL $80,78 \pm 2,053\%$, n=21

FZD₆ wt experiment after CL $70,69 \pm 2,416\%$, n=31

Statistical methods have been added to all figure legends and are better defined in the methods section/statistical analysis.

Moreover, we did not include the direct, statistical comparison of dcFRAP data obtained from FZD₆-FZD₆ vs FZD₆-CB₁ experiments as suggested by the reviewer. The statistical comparison of experiments that are based on different red/green expression ratios appeared difficult and we think it cannot really be justified. On the other hand, the stoichiometer allowed quantification of the ratios in the different experimental setups arguing that the red-green ratio is larger in the case of FZD₆-CB₁ (see point 1 in the general introduction). Since no interaction between FZD₆ and CB₁ could be observed despite a larger red-green ratio, we can safely conclude that the interaction between FZD₆ and CB₁ is negligible.

2. REVIEWER STATED: please show FRAP curves that underlie data from Fig 4a, at least those for the time points 0 and 5 min (GFP only would be sufficient). The reader wants to see what is the effect of agonist on the receptor interactions.

RESPONSE: That was a very good suggestion and we have included dcFRAP curves for the time points CL (0 min), 5, 15, and 20 min post WNT-5A stimulation (see Fig. 5b).

3. REVIEWER STATED: Why are the FCCS blots in Fig 1 g and 4 b,c are blotted in a different way?

The FCCS data presentation has been changed substantially and the labelling of the graphs is now consistent.

4. REVIEWER STATED: The negative effect of TM5 expression on P-ERK1/2 (Fig.6e,f) is inconclusive. If the effect would have been the opposite, the authors would have also claimed that it would support their claim. I would rather prefer to show it in the supplements.

RESPONSE: This comment is duly noted and it is also in agreement with the criticism of reviewer #2. Based on the experimental challenges related to proving the negative allosteric modulator (NAM) nature of TM5 bound to FZD₆ (i.e. ligand effect on potency, efficacy or affinity of WNTs), we have decided to collect the data with the TM1/TM5 minigenes in the supplementary Fig 7. Nevertheless, there is a clear effect of TM5 on FZD₆ dimerization and its signaling output towards ERK1/2 which justifies its inclusion in the study. In addition, the new biochemical analysis of the constitutive activity of the dimer mutant FZD₆ supports the idea that the monomeric FZD₆ is the active species.

5. REVIEWER STATED: Any explanation why the effect on crosslinking on FZD-GFP FRAP curves seem to be larger in Fig. 5 compared to Fig. 1?

RESPONSE: The validation of the experiments using the mCherry-TM-GFP stoichiometer revealed that the expression ratio of V5-FZD₆-mCherry and FZD₆-GFP in the experiments with panDVL siRNA and DVL2 overexpression were in the range of 5:1. This ratio is the underlying cause for the larger Δ between uncrosslinked and crosslinked FRAP curves. In addition, DVL overexpression affects the shape of the FZD recovery curves possibly due to DIX-domain polymerization of DVL thereby forming larger complexes associated with FZD.

We did not make any changes in the manuscript directly related to this comment.

6. REVIEWER STATED: Last but not least important: The authors did not show whether the dimerization deficient mutant is less or more active in the presence of WNT5a. This is very critical

in respect to interpretation of the importance of dimerization for the signaling phenotype of the mutant. Since there are numerous mutations in GPCRs that lead to an increase in basal activity (often accompanied by higher agonist affinity) it is important to know whether the mutation would lead to an increase in basal activity at the cost of the amplitude of the agonist induced activity.

RESPONSE: Please see major change #2 above. Employing the porcupine inhibitor C59, we provide clear evidence that the dimer mutant FZD₆, but not the wt, shows constitutive, ligand-independent activity to the ERK1/2 pathway.

REVIEWER STATED: Minor comments:

There are a couple of mistakes in the manuscript. Please reedit it for proper spelling: i.e.

Figure 5 legends: 1.8 asbling (assembling)

Results page 10: 210-214: asbly → assembly further, the % numbers are not explained: At first I thought the authors measure 63% assembly (dimer formation), but that is not what they mean.

RESPONSE: Spelling mistakes were corrected. Perhaps this was missed by the referee, but the percentages in the dcFRAP experiments represent the mobile fraction of the receptors in the measured ROI, i.e. the maximal fluorescence intensity of the protein of interest that would recover after photobleaching. In order to clarify and to define the term mobile fraction, we have included the following sentence when introducing the dcFRAP technique:

“Surface immobilization of the V5-FZD₆-mCherry was achieved with a biotinylated anti-V5 antibody and avidin and routinely resulted in a dramatic reduction in the recoverable fluorescence of V5-FZD₆-mCherry defining the mobile fraction of the protein of interest.”

REVIEWER STATED: „did not exhibit a reduction in the fluorescence recovery of the CB1-GFP mobile fraction upon crosslinking „

RESPONSE: This sentence was changed to “did not exhibit a reduction in the mobile fraction of the CB₁-GFP upon crosslinking”

REVIEWER STATED: Line 223 „receptor returned to values rebling pre-stimulation" (page 10)

RESPONSE: Corrected

G: The references give appropriate credit to previous work

H: The abstract /summary is well written and clear. Again the claims set out here need some validation as mentioned above

Reviewer #2 (Remarks to the Author):

REVIEWER STATED: This manuscript presents an interesting set of data about interactions of Frizzled6 in the membrane and apparent changes in response to the agonist WNT5a. The overall interpretation of the data is that the receptor is normally an inactive dimer that binds 1 DVL and 1 G protein and that this dimer separates to produce the active species that activates G protein and leads to ERK phosphorylation.

RESPONSE: This statement is meant to present our starting and working hypothesis by building on what we published in 2014 in Kilander et al., FASEB J (Kilander et al., 2014). In the current study, we address the dimer interface of FZD₆ and the independence of dimerization from DVL and G proteins. We do not make conclusions about receptor complex composition (receptor:DVL:G protein). We have reworded the passages and hope that the hypothetical nature of the statement is more obvious.

REVIEWER STATED: On the one hand the story hangs together and is potentially an intriguing advance. On the other hand, most of the conclusions are built on data that show extremely small and in the case of the FCCS, variable and inadequately powered experiments. The language in the paper is problematic, with grammatical and syntax errors, neologisms or repeated typographical errors like "asbly" and "rebling" that occur throughout. The language problems bleed, at times, into an overexaggeration of the results, without the presentation of alternative explanations or caveats, or in some cases necessary controls.

RESPONSE: We thank the reviewer for this criticism. In response, we have made a serious attempt to refrain from drawing conclusions which are too strong and not experimentally validated.

A fairly trivial but early example of the overreach is the sentence on page 5: "Since simultaneous interaction of DVL and G proteins with one FZD6 molecule appears competitive, we propose a dimeric FZD quaternary structure, in which one protomer interacts with DVL and the other one with the heterotrimeric G protein." This is fine to propose, but it is purportedly based on data about "one" molecule of FZD6 - how do the authors know what happens to single molecules of FZD6?

These are conclusions drawn from ensemble FRAP data with very small changes in recovery times, done with what appears to be minimal attention to stoichiometry. I am fine with a hypothesis drawn from the data but not to argue it is based on knowing something about a single molecule of FZD6 is careless language that can mislead.

RESPONSE: As already mentioned above, this statement is used to present our working hypothesis. We do not make conclusions about receptor-DVL-G protein complex stoichiometry based on the presented data. In order to clarify this, we have reworded that particular passage.

REVIEWER STATED: Taken together these problems in language and argument lead to a sense of overreach that pervades the manuscript in which very specific molecular models are presented as established, when in fact they are an interpretation of the data, which are in many cases not sufficiently explored.

RESPONSE: The reviewer is completely correct that we should not overstate and make the impression that we have investigated with single molecule resolution while applying population-based assays. This passage has been reworded.

REVIEWER STATED: First the FRAP data. The authors achieve efficient immobilization of the target protomer with their crosslinking method, down to about 25% of control. The associated change in the second protomer is vastly smaller, from 74% to 64%, a 10% change instead of a 75% change. The nonsignificant CB1 control in SF1a shows a change of perhaps 5%. The differences in Figure 2 between what is significant and not are tiny and this is very worrisome - 2i shows a dimer control that is reportedly still a dimer whereas 2k shows that TM5 disrupts dimerization - yet the results seem essentially the same in value although statistically consistent with the interpretation. My interpretation, rather than a stable dimer interface is that this at best suggests a minor transient interaction that might slow the partner but in no way truly immobilizes it. (Compare with the much

greater impact on the interaction with DVL shown in the SI.) If the dimer is stable then increasing the ratio of protomer A to protomer B should cause more and more of B to be associated with A and should enhance the effect. The only attention given to stoichiometry is the statement that equal amounts of receptor were necessary to see this, which doesn't seem consistent with the interpretation. Both protomers are theoretically the same except for their tags, so the more the A in the membrane, the more B should immobilize, and the more the B less B should immobilize. These controls seem to me to be essential. Another control would be to force interaction between the two protomers by chemical crosslinking or by the rapamycin system and see what happens. Efficient crosslinking should mimic a stable interface and lead to a much bigger effect of immobilizing A.

RESPONSE:

Signal amplitude: The assumption that we would expect 75% green immobilized receptors when we have immobilized 75% of the red receptors is not correct. In this experimental context, dimers can be red/red, green/green, red/green or green/red. Hypothetically speaking, this would provide 50% mixed color dimers at most. This population of mixed color dimers can be assessed by dcFRAP because they consist of one protomer that is immobilized and the other one that is detectable by GFP fluorescence. That leaves us with a maximal theoretical reduction in the mobile fraction of 37.5% in an experiment where we have 75% CL-induced immobilization of red receptors.

For the further criticism, we refer the reviewer to the listed major change #1 (see above), the introduction of the mCherry-TM-GFP stoichiometer, which we believe turned out to be extremely useful in responding to the reviewer's concerns. This tool allowed us to determine mCherry:GFP ratios and thereby we can now make a valid statement about stoichiometry of the red vs green population of receptors. In addition, the use of the stoichiometer construct allowed us to indirectly validate the statement of the reviewer arguing about changing red/green ratios to assess the effect on dcFRAP efficiency. For example, the dcFRAP experiments shown in Fig. 1 are based on a red/green ratio of 1.58:1, whereas the experiments in Fig. 4 are based on a red/green ratio of 5:1, a difference that in part caused a larger Δ in Fig. 5.

In addition, the comparison with the FZD₆-DVL interaction is tricky. DVL is polymerizing through its DIX domain at the same time as it is interacting with FZDs (forming so called signalosomes). Thus, coexpression of FZD₆ and DVL can reduce the mobile fraction of the FZD while simultaneously affecting the shape of the curve since it provides an additional degree of "crosslinking" on the intracellular side of the plasma membrane. Thus, the interaction mode of protomers in a GPCR dimer and FZD-DVL cannot be compared with regard to the level of CL-induced decrease in the mobile fraction.

REVIEWER STATED: The FCCS data appear to be underpowered and problematic. The cell numbers are not sufficient for the analysis, as can be judged by the enormous spread of the data in Figure 1g and the quite substantial cross correlation of proteins that based on FRAP are said to not interact. This is not consistent with the methods (line 580) stating that cross talk is not an issue.

RESPONSE: The presentation of the FCCS data in Fig. 1 has been adjusted and substantially redone. Details are described in the methods section. The analysis is based on hundreds of measurements from 6 independent experiments. Now, we compare the FCCS experimental data obtained from FZD₆/FZD₆ and FZD₆/CB₁ transfected cells to theoretical models of dimerization arguing that the FCCS data of FZD₆/FZD₆ transfected cells shows the best fit with a model that assumes FZD₆ dimerization when CB₁ is monomeric. Please see the new Fig. 1j and table in supplementary figure 1b.

REVIEWER STATED: The authors don't explain the y-axis in Fig 1g, which I presume it is a percent (i.e. they multiply the ratio in the rest of the FCCS y axes by 100), but it's not consistent with Fig 2, so this is difficult to interpret. The example in Fig 1h looks like the baseline is floating and the model function fit wouldn't be very good. The model function fits should be shown throughout.

RESPONSE: The original presentation of the data in Fig 1g was removed and replaced by Fig. 1i, j. The model function fits are now included throughout.

REVIEWER STATED: The cross-correlation amplitudes in Fig 2h aren't helpful without the autocorrelation functions. We need to see how they compare to their respective autocorrelation functions, not just the absolute comparison. In Fig 2g, the N is 6 cells, which is not enough based on the variation shown. The data for WT diverge greatly from Fig 1. Although the y axes problem noted above precludes certainty about this.

RESPONSE: We have replaced the FCCS analysis of the wt vs dimer mutant FZD₆ in Fig. 2 with FCS using the GFP-tagged receptor constructs. The advantage here, compared to the previous FCCS setup, is that the experiments are performed with wt and mutant receptors only and not with a mixture of red/green wt and red/green wt/dimer mutant. The difference in τ_D presented in the new Fig. 2i is indicative of a smaller and faster receptor complex in the case of the dimer mutant compared to wt-FZD₆. This experimental design avoids receptor populations that are still interacting (e.g. weakened interaction in a wt/dimer mutant complex or presence of wt/wt dimers) and supports the hypothesis that wt receptor complexes are larger and diffusing slower compared to the dimer mutant.

REVIEWER STATED: The comparison in 4d is likely more legitimate because the same cell should have the same population over time, although, this should still be shown. The time-dependence is quite intriguing and consistent with the interpretation of the FRAP data, but we still need to see the Nr and Ng values over time to know if receptors are being internalized or photobleached during the long time measurement.

RESPONSE: In order to address the reviewer's criticism that receptor trafficking or photobleaching could at least in part explain the changes observed by FCCS (presentation of Ngr/Nr over time after WNT stimulation), we have complemented the whisker plot (Fig. 5c) with a secondary axis plotting the particle number of Nr and Ng over time. The data for Nr and Ng over time were normalized to the first time point assessed. The increase in particle number observed in Nr and Ng coincides with the dip in Ngr/Nr. Particle numbers Nr and Ng at baseline and during the late part of the measurements are similar (especially for Ng, where the fluorophore is less sensitive to photobleaching) indicating that receptor trafficking is minimal over the time measured. This is further corroborated by confocal imaging of SNAP-FZD₆ (in comparison with SNAP- β_2 -AR) after agonist stimulation over time (Fig. 5f).

REVIEWER STATED: The methods contain multiple grammatical errors. The authors call a GFP-only experiment a 'negative control', but don't show any of the data. Line 574 - I have no idea what the authors mean by "were estimated by visual inspection." If taken literally, this is problematic.

RESPONSE: This section has been corrected and reworded. Crosstalk correction procedure is now described in the methods section.

REVIEWER STATED: Conclusions were drawn in a previous manuscript about the mutant 511C not interacting with G proteins based on FRAP data. In this manuscript we learn that 515C is not expressed at the plasma membrane but rather in lysosomes and other intracellular organelles. Thus arguing mechanistically from these data about G protein coupling is dangerous.

RESPONSE: In the original paper (Fröjmark et al 2011 Am J Hum Gen), where the FZD₆-R511C nail dysplasia mutant is described, we quantified the relative membrane expression of this mutant. Despite the fact that most of the mutant receptor is internalized, some cells (<50% adding up those that show membranous and membrane/vesicular distribution) actually show membrane expression. Those are the cells that were chosen for the dcFRAP measurements. Naturally, those measurements would not be possible if the receptor would not be colocalized with the G protein in the membrane. Since the dcFRAP assay is based on single cell analysis and because only cells were chosen that coexpress receptor and G protein in the membrane, the conclusion about G protein interaction, or better the lack thereof, is valid.

For clarity, we provide here the graph of the quantification of wt-FZD₆-GFP vs FZD₆-R511C-GFP (FZD₆-missense) from Fröjmark et al. (Fröjmark et al., 2011)

REVIEWER STATED: The authors write: "Since neither R511C nor the dimer mutant of FZD6 were able to assemble with heterotrimeric G proteins, we collectively conclude that neither DVL nor G protein binding are required for the formation of an inactive-state FZD6 dimer (Fig. 5g)." ("We collectively" means all the authors concluded on this together...)

RESPONSE: Spelling errors were corrected. Sentence was reworded to avoid "collectively."

REVIEWER STATED: First we have the problem with 511C mentioned above.

RESPONSE: See explanation above regarding a minor proportion of FZD₆-R511C receptors being localized in the plasma membrane enabling the dcFRAP assay with fluorescently tagged G proteins.

REVIEWER STATED: Second, a major finding of the paper is that the mutant that apparently cannot assemble into dimers still functions, in fact perhaps better than WT, to activate ERK, and since this effector pathway is reported to be partially sensitive to PTX, then the mutant does assemble with G protein, just not in a way that can be measured by the FRAP experiments. Statements like this are made like logical truths but when I try to break them down and understand what they mean in detail, problems like this arise.

RESPONSE: I think that the misunderstanding or the problem here might be semantic or at least a common issue with nomenclature regarding G protein assembly, precoupling and G protein coupling. In our experimental setup, we have previously argued that what we measure with dcFRAP when we investigate FZD and G protein interaction is in fact an "inactive state complex" similar to what Nevin Lambert and others have described for the M3 receptor using the same methodology (Qin et al Nat Chem Biol 2011). This should be clearly distinguished from receptor-G protein coupling, which resembles a state that is basically caught in the "Kobilka structure" of an adrenergic receptor coupled to the G_{αs} protein in the presence of an agonist and the absence of a guanine nucleotide (only the GDP/GTP-free G protein couples to the receptor) – Rasmussen et al., 2011, Science. Less well defined concepts are "precoupling" or "assembly". These states refer to the differences between a G protein that is in association with the receptor in a GDP bound state, but in the absence of agonist. Upon agonist binding, the G protein couples and dissociates. The other extreme is the so-called "collision coupling" where agonist-bound receptors only establish contact and couple to the receptor in a kiss-and-run fashion. The dcFRAP methodology in live cell imaging (where G proteins are mostly GDP/GTP bound) only allows us to detect receptor-G protein assembly and not collision coupling. Based on the findings that agonist stimulation dissociates the FZD-G protein complex, that we cannot detect FZD₆-dimer mutant G protein assembly and the fact that the dimer mutant actively induces downstream signaling in the absence of ligand, we conclude that the dimer mutant cannot establish an inactive state complex with the G protein. This is supported by the novel biochemical data provided in Fig 6, which more clearly indicate that the FZD₆ dimer mutant has a higher constitutive activity than the wild type. Thus, a constitutively active receptor is not – according to the ternary complex model – expected to bind the G protein in a way that is detectable by dcFRAP. We have added the reference to (De Lean et al., 1980), the initial description of the ternary complex model to the text.

REVIEWER STATED: The lack of significant movement at the interface of the molecular dynamics experiments is consistent with the interface being a reasonable one, but simulations of aggregate 100 ns are inadequate to speak to the stability of the interface. The length of simulation is not sufficient for sufficient evolution of a protein-protein interface, even if one wanted it to move. Thus, whether the interaction at the interface is transient or stable simply cannot be based on these data, which are again overinterpreted in my opinion as the general sense is that the authors propose a stable dimer until agonist comes, except when they want to increase the monomer concentration (see below) when they talk about an equilibrium suggesting that it is a transient interface that exchanges.

RESPONSE: The reviewer is completely correct that we cannot draw any conclusions from the molecular dynamics simulations about dimer stability. Those statements have been corrected. Instead, we argue that the MD simulations validate the proposed dimer interface and argue that the interface is a likely one. In general, we avoid the term "stable dimer" to avoid implying that the dimers are static. Rather, we argue that FZD₆, in the absence of WNT-5A, presents a larger population of dynamic FZD₆ dimers than after WNT-5A stimulation. In addition, we have extended the molecular dynamics simulation to 100 ns unrestrained simulation time totaling 300 ns (100 ns times 3) to further validate the feasibility of the dimer interface.

REVIEWER STATED: "Interestingly, the kinetics of WNT-5A-induced ERK1/2 phosphorylation matched the time course of FZD6 de- and re-dimerization, suggesting a role for dimerization in signaling." Suggesting a "possible" role would be less of a reach. A similar time course does not a mechanistic connection make.

RESPONSE: We include "possible"

REVIEWER STATED: "...confirming loss-off-function with gain off-function data (Fig. 6d)." I know what the authors mean but the sentence is problematic.

RESPONSE: In order to clarify the statement, we have reworded the passage to: Along the same line, CRISPRa-induced enhancement of FZD₆ expression in MLE-12 cells resulted in increased C-RAF and ERK1/2 phosphorylation confirming that regulation of FZD₆ protein levels relates to the degree of ERK1/2 phosphorylation (Fig. 6d).

REVIEWER STATED: "Furthermore, overexpression of FZD6 WT results in quantitatively more monomeric receptors in a dynamic dimer/monomer equilibrium leading to enhanced signaling to ERK1/2. Overexpression of the FZD6 dimer mutant amplifies the number of monomeric receptors with stronger signaling output as a consequence." Increasing expression of WT is very difficult to interpret. It could make more monomers in to a transient dimer model - although for most of the paper it sounds like the model is for a very stable interaction until agonist. But of course overexpression of WT should also increase dimers, so it is also consistent with true dimers signaling but the inability of a WT protomer bound to a TM5 minigene product to signal. Many interpretations are possible but the statements are broad and taken as clear evidence of the proposed solution, without consideration of the caveats or alternatives. The dimer data are interesting and do suggest that this construct can signal. It is unclear to me from the data whether it is sensitive to agonist or not or just constitutively active, so again multiple interpretations are possible and perhaps agonist response needs a dimer. When one really gets down to the details, the conclusions become less clear, even though the data are interesting. More careful examination of the alternatives would drive experiments that might allow these alternatives to be differentiated rather than stated.

RESPONSE: The reviewer is correct and we are particularly thankful for this comment. First of all, we moved the dcFRAP data and the P-ERK1/2 with TM1 and TM5 minigenes to the supplement,

which was in part also suggested by reviewer 1. Based on the experimental challenges related to proving the negative allosteric modulator (NAM) nature of TM5 bound to a FZD₆ (i.e. ligand effect on potency, efficacy or affinity of WNTs), we have decided to collect the data with the TM1/TM5 minigenes in Supplementary Fig 7. Nevertheless, there is a clear effect of TM5 on FZD₆ dimerization and its signaling output towards ERK1/2 which justifies its inclusion in the study.

Most importantly, we have added experiments to more clearly define that the dimer mutant – in contrast to the wild type receptor – does not depend on WNTs for signal initiation to ERK1/2. Due to the general lack of pharmacological tools in the Frizzled field, we have used an indirect approach employing a porcupine inhibitor (C59), which abrogates endogenous WNT secretion in cells. For more details, see major changes (point 2) at the beginning of the rebuttal letter.

REVIEWER STATED: More examples of complicated and misleading sentences follow:

"Moreover, the FZD6 dimer can form independently of DVL or heterotrimeric G proteins (Fig. 5)." As noted above dead intracellular dimers or aggregates can form without the ability to show G protein interaction by FRAP and clearly the G protein interaction is missed with the dimer deficient mutant that signals, so this entire line of argument seems problematic and unnecessary.

RESPONSE: The sentence has been removed.

REVIEWER STATED: "It is likely, but so far speculative, that agonist-induced dedimerization and G protein dissociation occur simultaneously, whereas the FZD6-DVL interaction could be static." Why is this likely and not just speculative? It requires data or a logical flow.

RESPONSE: In agreement with the reviewer's criticism, we have completely removed this statement.

REVIEWER STATED: As noted above, it isn't even clear to me that the monomer is the agonist responsive species. How do we know if internalization or some other change in response to agonist is giving a confound - as noted above the FCCS data is wanting controls here as well to show that the autocorrelation is not altered.

RESPONSE: In order to exclude that agonist-induced receptor internalization could be the source of the transient decrease in cross correlation, we (i) added the values for Nr and Ng into Fig. 5c and (ii) we performed additional experiments where we monitor fluorescently labelled FZD₆ after WNT-5A stimulation over the time course that we performed the FCCS experiments. Similar to what we had observed earlier ((Kilander et al., 2014)– supplemental figure 2I) we could not observe an agonist-induced increase in FZD₆ internalization.

The other point that the reviewer raises about the WNT-sensitive receptor species is highly interesting. Unfortunately, the necessary tools to study this phenomenon are lacking. What we speculate is that the dimer is the agonist-interpreting species inducing dedimerization. What happens then will require many more experiments and the development of new tools. Are 1 or 2 WNTs required per dimer? Is the monomer still bound to an agonist or already desensitized? What is the residence time of WNTs? Does the WNT dissociate when the receptor complex reassociates? At what point of the dedimerization/reassociation cycle does the G protein interact with the receptor in a guanine nucleotide free state (when does it actually couple?)? And at what state does it dissociate? Are dissociation of G protein and dimer parallel processes or is the dimer dissociated by the coupling of the G protein and the release of GDP?

REVIEWER STATED: "This is in agreement with several lines of evidence supporting that monomeric rather than dimeric GPCRs present the true active conformation (Fig. 6)4,41,44." The one true conformation is not a helpful concept. These papers show or infer that monomers CAN signal. They say nothing about what happens in the membrane and being monomeric isn't really a conformation anyway.

RESPONSE: The sentence has been reworded: This is in agreement with several lines of evidence supporting that the active conformation observed in GPCRs is associated with the monomeric rather than dimeric receptor species.

REVIEWER STATED: "Also, the phenomenon of dynamic and agonist-sensitive dimerization might not be restricted to Class F receptors, but could provide a more general concept of GPCR signal transduction." This comes after discussion about the differences and how Family A are all transient and not shown to respond to agonist (except this is contradicted by the discussion on alpha 2C) so unclear what is being said. The area of Family A dimerization of GPCRs is controversial and hotly debated and general statements like this are not helpful.

RESPONSE: We clearly understand the reviewer's criticism. However, we think that one plausible conclusion from our study is that dimerizing GPCRs could undergo ligand-dependent dynamics as a prerequisite of signal initiation. In order to stimulate further research in this direction, we feel that this speculation is justified at this place, especially with regard to recent data on complex dynamics in Class B and Class C receptors as mentioned in the discussion. We clearly label this statement as speculation and decided to leave it as it was.

REVIEWER STATED: There was evidence previously for dimerization of other FZDs and that this dimerization is essential to get to the surface. For FZD6 the current data suggest that this is not the case as the dimer deficient mutant gets to the surface and signals, at least constitutively (see above on the need to assess its response to agonist). Thus, there is likely to be a diversity of mechanisms even for different FZDs. What the field needs is clear, logical, methodologically sound steps towards consensus. The authors don't discuss this difference of the FZD literature with their current inferences about FZD6 but instead make grand sweeping and largely uninterpretable claims about a general concept of GPCR signal transduction.

RESPONSE: The reviewer is referring to the paper by (Kaykas et al., 2004), which we also cite in the manuscript. However, the reviewer apparently misinterpreted the conclusion of that study since the conclusion is not that FZD₄ dimerization is required to reach the surface, but rather that oligomerization occurs in the ER. The FEVR FZD₄ mutant appears to be structurally impaired and unable to reach the cell surface. This results in sequestration of the coexpressed wt-FZD₄ in the ER through receptor-receptor interaction thereby preventing surface expression of the wt and blocking FZD₄-mediated signaling. Thus, the reason that we do not discuss this differences of the FZD literature is that we cannot argue – based on our data - if FZD₆ dimers are formed in the ER or at the plasma membrane and we cannot argue either if the dimerization modes of different FZDs are similar or not. The mutant FZD₄ reported in Kaykas et al., is still capable of forming dimers, but is stuck in the ER.

Since our manuscript addresses a completely different aspect of receptor dimerization, namely, a dynamic dimerization in the plasma membrane compared to a pathological mutated receptor that is trapped in the ER, we cannot compare the phenomena studied in these two papers. However, we refer (as we did in the original submission) to the finding from (Kaykas et al., 2004) showing that FZD dimerization is occurring through interaction of the heptahelical domain and independent of the CRD.

References

- De Lean A, Stadel JM and Lefkowitz RJ (1980) A ternary complex model explains the agonist-specific binding properties of the adenylate cyclase-coupled beta-adrenergic receptor. *J Biol Chem* **255**(15): 7108-7117.
- Frojmark AS, Schuster J, Sobol M, Entesarian M, Kilander MB, Gabrikova D, Nawaz S, Baig SM, Schulte G, Klar J and Dahl N (2011) Mutations in Frizzled 6 cause isolated autosomal-recessive nail dysplasia. *Am J Hum Genet* **88**(6): 852-860.
- Kaykas A, Yang-Snyder J, Heroux M, Shah KV, Bouvier M and Moon RT (2004) Mutant Frizzled 4 associated with vitreoretinopathy traps wild-type Frizzled in the endoplasmic reticulum by oligomerization. *Nat Cell Biol* **6**(1): 52-58.
- Kilander MB, Petersen J, Andressen KW, Ganji RS, Levy FO, Schuster J, Dahl N, Bryja V and Schulte G (2014) Disheveled regulates precoupling of heterotrimeric G proteins to Frizzled 6. *FASEB J*.

- Qin K, Dong C, Wu G and Lambert NA (2011) Inactive-state preassembly of G(q)-coupled receptors and G(q) heterotrimers. *Nat Chem Biol* **7**(10): 740-747.
- Qin K, Sethi PR and Lambert NA (2008) Abundance and stability of complexes containing inactive G protein-coupled receptors and G proteins. *FASEB J* **22**(8): 2920-2927.
- Sprague BL and McNally JG (2005) FRAP analysis of binding: proper and fitting. *Trends Cell Biol* **15**(2): 84-91.

Reviewers' comments:

Reviewer #1 (Remarks to the Author):

The revised manuscript of Petersen et al. entitled: „Agonist-induced dimer dissociation as a macromolecular step in G protein-coupled receptor signaling” has improved by trying to address the criticism of the reviewers. For instance, an important improvement resulted from using a stoichiometer, which helped to define relative expression levels. In the following I will give my opinion on how the authors improved the manuscript and where further improvements are needed.

1. (previous review): It is well taken, that the authors consider the kinetics of the FRAP curves too complicated. However it is not clear to me why they now choose a different time span to analyze their data, since latest at 50 s after bleaching the curves run parallel. Considering the estimated dimer fraction, the authors estimate it to be around 33 %. Considering the relative expression level of red versus green receptors, this number seems even a bit overestimated. Therefore the question that I raised remains: Why do the authors not conclude that the majority of these receptors do not form (stable) dimers? Furthermore, the dcFRAP curves suggest, that there is no major fraction of transiently interacting receptors detectable. How does these results from the FRAP experiments fit to the conclusion of the cross correlation spectroscopy studies, that all FZD6-R form dimers?

2. The argument brought forward by the authors, that the conditions between control (CB1-R) and test conditions (FZD6) are too dissimilar to allow for a statistical comparison is not well taken. Even if the conditions were somewhat different a direct comparison to controls is necessary. This is always important, however if we see only small differences in the data (FRAP curves), it becomes particularly important to measure and analyze data carefully. The conclusion that FZD receptors exhibit a specific oligomerization is not justified based on Fig. 1b-g. There must be a direct statistical comparison between control (fig. 1g) and test (Fig. 1d) condition in order to conclude that there is a specific oligomerization between FZD6 receptors.

After careful looking at the figures I noticed that the identical set of data for the FZD6-GFP recovery curve in the absence of crosslinking is now been shown not only in Fig. 1c but also in Fig. 4b and here even for all different time points. I feel that this is not appropriate, since (a) the figure now suggests that GFP recovery has been measured for both crosslinked conditions and non-crosslinked conditions in dependence of agonist exposure, which is actually not true. (b) Showing the same data twice in two different figures must be justified and mentioned. It is a good lab practice to measure controls always in parallel to test conditions in order to rule out influence of unknown factors. Again to ensure the existence of relatively small differences this is quite important. Along this line it would actually be important to measure controls (no crosslinking) also after agonist exposure (Fig. 4b) in order to rule out any time dependent or agonist dependent variation in the lateral mobility of these receptors.

Reviewer #3 (Remarks to the Author):

This is a very interesting manuscript that will be of broad interest to scientists in the field of GPCR research. The authors have gone a long way to addressing the comments raised by reviewers. They have added substantial new data and undertaken new analyses of the data obtained. I think the new controls that have now been included have substantially strengthened the manuscript and I am happy to recommend publication.

Rebuttal letter

Reviewers' comments:

Reviewer #1 (Remarks to the Author):

The revised manuscript of Petersen et al. entitled: „Agonist-induced dimer dissociation as a macromolecular step in G protein-coupled receptor signaling” has improved by trying to address the criticism of the reviewers. For instance, an important improvement resulted from using a stoichiometer, which helped to define relative expression levels. In the following I will give my opinion on how the authors improved the manuscript and where further improvements are needed.

Response: We appreciate the positive comments about our revised work.

1. (previous review): It is well taken, that the authors consider the kinetics of the FRAP curves too complicated. However it is not clear to me why they now choose a different time span to analyze their data, since latest at 50 s after bleaching the curves run parallel.

Response: Based on the work of Sprague and McNally (Sprague and McNally, 2005) and along the same line as previous publications employing the same variant of the method (Qin et al., 2008; Dorsch et al., 2009), we chose to analyse the time interval consisting of the last measurements in order to present convincing data about binding over diffusion. While it is true that most uncrosslinked and crosslinked curves run parallel, both conditions (depending on the proteins in question) exhibit varying degrees of diffusion and binding which, as the referee has acknowledged, are too complicated to analyse and perhaps beyond the scope of this manuscript. Therefore, for the sake of consistency and to be in line with other important publications in the field of dimerization (Dorsch et al., 2009), we consider the later time interval.

[redacted]

Considering the estimated dimer fraction, the authors estimate it to be around 33 %. Considering the relative expression level of red versus green receptors, this number seems even a bit overestimated. Therefore the question that I raised remains: Why do the authors not conclude that the majority of these receptors do not form (stable) dimers? Furthermore, the dcFRAP curves suggest, that there is no major fraction of transiently interacting receptors detectable. How does these results from the FRAP experiments fit to the conclusion of the cross correlation spectroscopy studies, that all FZD6-R form dimers?

Response: In the previous rebuttal letter, we provided hypothetical values for mobile fractions in the dcFRAP assay, which might have been misinterpreted by the referee as being representative of our data when in fact our experimental results point to values lower than 33%. We agree completely with

the referee that the degree of dimerization that we observe is less than the theoretical maximum of the assay. Extrapolating from experiments with varying relative expression ratios of mCherry to GFP, we most likely observe a dimeric fraction around 25% when the ratio of mCherry to GFP is 1:1. This is in line with FCCS data presented in Fig. 5c. To be clear, the FCCS data presented in Fig. 1j presents our findings against a theoretical model where we assume that FZD₆ dimerizes. As one can see, the experimental data follow the theoretical data along different expression levels of green and red. This is clearly not the case for FZD₆ and CB₁ supporting our model and the use of CB₁ as a negative control. As stated above, we argue that the total population of FZDs are in a dynamic equilibrium of monomers and dimers. The monomer–dimer equilibrium can be shifted by the presence of the endogenous ligand (WNT), which are secreted by many cells including HEK293 cells (see Fig. 6). Other processes which have not yet been defined could surely have an impact on the receptor heterogeneity and the degree of dimerization such as posttranslational modifications, compartmentalization and interaction with other proteins. In summary, our data add to the existing concepts of GPCR dimerization as being either transient or stable because our findings support a novel model that is based on a change in affinity between the protomers depending on presence of the agonist. Thus, we acknowledge the coexistence of dimers and monomers where monomers might exceed the concentration of dimers and dimers might exceed the concentration of monomers depending on the surrounding milieu (agonist presence). The fact that we can have a shift in favor of one population or the other is a finding, which is unprecedented and has important implications for the field of GPCR dimerization (with the caveat that general applicability needs to be validated in future experiments).

In order to be more explicit regarding the dynamic nature of the FZD₆ dimers, we introduced the following passage on p18/line 415:

“Our findings extend beyond the pre-existing concepts of constitutive, transient or stable GPCR dimerization by introducing the idea that FZD₆ dimers dissociate and re-associate in a ligand- and receptor-activity-dependent manner.”

2. The argument brought forward by the authors, that the conditions between control (CB1-R) and test conditions (FZD6) are too dissimilar to allow for a statistical comparison is not well taken. Even if the conditions were somewhat different a direct comparison to controls is necessary. This is always important, however if we see only small differences in the data (FRAP curves), it becomes particularly important to measure and analyze data carefully. The conclusion that FZD receptors exhibit a specific oligomerization is not justified based on Fig. 1b-g. There must be a direct statistical comparison between control (fig. 1g) and test (Fig. 1d) condition in order to conclude that there is a specific oligomerization between FZD6 receptors.

Response: We thank the reviewer for the comment and agree that the statistical comparison of the CL-induced change in the mobile fraction of GFP in FZD₆-FZD₆ and FZD₆-CB₁ expressing cells adds clarity to the data presentation. We have taken two approaches to statistically assess the differences between the crosslinking-induced reduction in the mobile fraction of the GFP-tagged constructs in the conditions V5-FZD₆-mCherry/FZD₆-GFP and V5-FZD₆-mCherry/CB₁-GFP. In discussion with the editor, we decided to present the direct comparison of the relevant data points from Fig. 1d, g (third and fourth columns of the figures) in Fig. S1b. In order to avoid to use of the same data sets from Fig 1d, g in another panel in Fig. 1, we show the bar graph and ANOVA comparison as a new panel in Supplementary Fig. 1a.

We have added the following sentence to the results section:

“Direct statistical comparison (one-way ANOVA) of the CL-induced differences in the mobile fraction of the respective GFP constructs in the conditions V5-FZD₆-mCherry/FZD₆-GFP and V5-FZD₆-mCherry/CB₁-GFP indicate that the observed interactions are specific (Supplementary Fig. 1a).”

Furthermore, we analyzed the differences in the changes in mobile fraction as the average of the mobile fraction from uncrosslinked experiments and used that to transform the data ($Y=Y-K$) in order to compare FZD₆/FZD₆ and FZD₆/CB₁. The data sets were subjected to statistical analysis by two-tailed t-test. We decided to only present this figure here in the rebuttal letter as a further evidence of the actual difference between the ctrl and experimental condition:

After careful looking at the figures I noticed that the identical set of data for the FZD₆-GFP recovery curve in the absence of crosslinking is now been shown not only in Fig. 1c but also in Fig. 4b and here even for all different time points. I feel that this is not appropriate, since (a) the figure now suggests that GFP recovery has been measured for both crosslinked conditions and non-crosslinked conditions in dependence of agonist exposure, which is actually not true. (b) Showing the same data twice in two different figures must be justified and mentioned. It is a good lab practice to measure controls always in parallel to test conditions in order to rule out influence of unknown factors. Again to ensure the existence of relatively small differences this is quite important. Along this line it would actually be important to measure controls (no crosslinking) also after agonist exposure (Fig. 4b) in order to rule out any time dependent or agonist dependent variation in the lateral mobility of these receptors.

Response: The referee is absolutely correct in that this data presentation (Fig. 4b) was not appropriate and we have since changed it. In the last revision round, we were focused on responding to the reviewer’s previous request to present all curves and did not pay sufficient attention to the details whilst putting the panel together. Showing the requested curves including data points and error bars in one graph, however, made it difficult to distinguish the individual data sets. In an attempt to make the data clearer, we decided to split the graphs using the same “prior CL” curves for comparison, which was surely a poor way to present the data. In order to present a less busy figure, we present the curves for unstimulated, 5 min and 20 min as a complement to the FCCS curves and to avoid the overcrowdedness with all data sets. We have chosen to exclude the curve showing FRAP in the absence of crosslinking as is common practice in previous publications by groups that have established the dcFRAP methodology (Qin et al., 2008; Dorsch et al., 2009). We adopted a clearer color scheme to improve the readability of the graph. In addition, we matched the color scheme in Fig. 5b (curves for dcFRAP) with the FCCS (cross-correlation changes over time with WNT stimulation) presented in Fig. 5d for clarity.

In order to avoid the presentation and comparison of the different FRAP curves in Fig. 5c with the uncrosslinked control presented in Fig 1 (experiments were done in parallel), we changed Fig. 5c as a means of presenting the effect of WNT stimulation on the change in the FZD₆-GFP mobile fraction only after crosslinking of V5-FZD₆-mCherry over time. In this way, we use the crosslinked/unstimulated values and the crosslinked/20 min time course as internal controls of the dynamic behavior of the FZD₆ dimer in 5c. As a further control that the WNT-induced increase and subsequent decrease in the mobile fraction of FZD₆-GFP under crosslinking conditions is indeed a consequence of dynamic dimerization, we would like to point out the differences in the uncrosslinked mobile fractions between V5-FZD₆-

mCherry and wt FZD₆-GFP (Fig. 1d) or dimer mutant FZD₆-GFP (Fig. 2e). When statistically comparing (unpaired t-test) the mobile fractions of these constructs, there is a significant difference only between V5-FZD₆-mCherry and dimer mutant FZD₆-GFP (P=0,033) and not V5-FZD₆-mCherry in comparison with either wt-FZD₆-GFP (P=0,364) or control dimer mutant FZD₆-GFP (P=0,308; Fig. 2g). Thus, we feel confident to conclude that the increase of the FZD₆-GFP mobile fraction with WNT stimulation argues in favor of dimer dissociation resembling the higher mobile fraction of the dimer mutant.

No technique is without its limitations and dcFRAP is no exception. Despite the advantage of being an affinity-based assay or “live-cell IP”, it suffers from a lack of kinetic resolution that other assays have (i.e. FRET or BRET) while boasting the ability to distinguish between non-interacting proteins which the two aforementioned techniques cannot do. That being said, dynamic dcFRAP measurements have never been published before to our knowledge in the study of GPCR dimerization and this presents some scientific hurdles that are not easy to overcome. Regarding the use of dcFRAP data shown in Fig 1 and Fig 5, we explicitly state and explain this in the figure text:

“Unstimulated (before and after CL) are from Fig. 1d and were performed in parallel with stimulation experiments.”

In addition to their time and labor intensive nature, these experiments are challenging with regard to the time pressure on finding appropriate cells to image, the acquisition of data as well as the time remaining for the next time point – not to mention balancing cell survival. Experiments are further compromised by stringent exclusion criteria (described in the methods section) that apply to the degree of CL and the displacement of cells during FRAP measurements. For this reason, we complemented our data set with FCCS, which does not require crosslinking and further supports the idea of agonist-induced dynamics in the dimer population.

We hope that our explanations clarify the issues raised and that our manuscript is now acceptable for publication.

Reviewer #3 (Remarks to the Author):

This is a very interesting manuscript that will be of broad interest to scientists in the field of GPCR research. The authors have gone a long way to addressing the comments raised by reviewers. They have added substantial new data and undertaken new analyses of the data obtained. I think the new controls that have now been included have substantially strengthened the manuscript and I am happy to recommend publication.

Response: We thank the reviewer for the positive comment and are happy that the efforts for the revision were appreciated and deemed to be sufficient.

REVIEWERS' COMMENTS:

Reviewer #1 (Remarks to the Author):

The authors have improved the manuscript considerably. All points raised in my last review have been addressed in a way that I agree with the other reviewers to finally support acceptance of the manuscript. I appreciate that the authors discussed with the editor data presentation and statistical analysis. There is clear evidence provided, that agonist binding to frizzled receptors alters the dimerization state of the receptor. This is important to the field since this is the first time that alteration of the dimerization state of a GPCR is linked to activation and signaling.

Rebuttal letter – final revision

Reviewers' comments:

Reviewer #1 (Remarks to the Author):

The authors have improved the manuscript considerably. All points raised in my last review have been addressed in a way that I agree with the other reviewers to finally support acceptance of the manuscript. I appreciate that the authors discussed with the editor data presentation and statistical analysis. There is clear evidence provided, that agonist binding to frizzled receptors alters the dimerization state of the receptor. This is important to the field since this is the first time that alteration of the dimerization state of a GPCR is linked to activation and signaling.

Response: We are especially thankful for and happy about this summary and acknowledgement of the relevance and novelty of our work.